Contributions of Trans-boundary Transport to Summertime Air Quality in Beijing, China

Jiarui Wu[1,3], Guohui Li[1*], Junji Cao[1*], Naifang Bei[2], Yichen Wang[1], Tian Feng[1,2], Rujin Huang[1], Suixin Liu[1], Qiang Zhang[4], and Xuexi Tie[1]

[1]Key Lab of Aerosol Chemistry and Physics, SKLLQG, Institute of Earth Environment, Chinese Academy of Sciences, Xi'an, China
[2]School of Human Settlements and Civil Engineering, Xi'an Jiaotong University, Xi'an, Shaanxi, China
[3]University of Chinese Academy of Science, Beijing, China
[4]Department of Environmental Sciences and Engineering, Tsinghua University, Beijing, China
[*]Correspondence to: Guohui Li (ligh@ieecas.cn) and Junji Cao (jjcao@ieecas.cn)

**Abstract**: In the present study, the WRF-CHEM model is used to evaluate the contributions of trans-boundary transport to the air quality in Beijing during a persistent air pollution episode from 5 to 14 July 2015 in Beijing-Tianjin-Hebei (BTH), China. Generally, the predicted temporal variations and spatial distributions of $PM_{2.5}$ (fine particulate matter), $O_3$ (ozone), and $NO_2$ are in good agreement with observations in BTH. The WRF-CHEM model also reproduces reasonably well the temporal variations of aerosol species compared to measurements in Beijing. The factor separation approach is employed to evaluate the contributions of trans-boundary transport of non-Beijing emissions to the $PM_{2.5}$ and $O_3$ levels in Beijing. On average, in the afternoon during the simulation episode, the local emissions contribute 22.4% to the $O_3$ level in Beijing, less than 36.6% from non-Beijing emissions. The $O_3$ concentrations in Beijing are decreased by 5.1% in the afternoon due to interactions between local and non-Beijing emissions. The non-Beijing emissions play a dominant role in the $PM_{2.5}$ level in Beijing, with a contribution of 61.5%, much higher than 13.7% from Beijing local emissions. The emission interactions between local and non-Beijing emissions enhance the $PM_{2.5}$ concentrations in Beijing, with a contribution of 5.9%. Therefore, the air quality in Beijing is generally determined by the trans-boundary transport of non-Beijing emissions during summertime, showing that the cooperation with neighboring provinces to mitigate pollutant emissions is a key for Beijing to improve air quality.

## 1    Introduction

Beijing, the capital of China, has become an environmentally stressed city due to growing population, increasing transportation activity, and city expansion (Parrish and Zhu, 2009). Beijing is situated in northeastern China, surrounded from the southwest to the northeast by the Taihang Mountains and the Yanshan Mountains and open to the North China Plain (NCP) in the south and east. Unfortunately, NCP has become one of the most polluted areas in China due to rapid industrialization and urbanization (Zhang et al., 2013). When south or east winds are prevalent in NCP, air pollutants originated from NCP are transported to Beijing and surrounding areas and subject to be accumulated due to the mountain blocking, causing heavy air pollution in Beijing (Long et al., 2016).

$PM_{2.5}$ (fine particulate matter) and $O_3$ (ozone) are considered to be the most serious air pollutants of concern in Beijing during summertime (e.g., Xie et al., 2015; Zheng et al., 2015; Chen et al., 2015; Wang et al., 2016). The mean summertime $PM_{2.5}$ mass concentration is about 80 $\mu g\ m^{-3}$ in 2013 (Li et al., 2015a), exceeding the second grade of National Ambient Air Quality Standards (NAAQS) in China and also higher than the average $PM_{2.5}$ concentration of 78.1 $\mu g\ m^{-3}$ during the period from 2004 to 2012 (Liu et al., 2015). During haze pollution events in summer 2014, the $PM_{2.5}$ concentration generally reaches 100 $\mu g\ m^{-3}$, and even exceeds 150 $\mu g\ m^{-3}$ in Beijing (Wang et al., 2016). An increasing $O_3$ trend has been observed in Beijing from 2002 to 2010 (Wang et al., 2012; Wang et al., 2013). The average maximum 1-h $O_3$ concentration has been reported to achieve 140 $\mu g\ m^{-3}$ during summertime of 2013 in Beijing (Wang et al., 2014a). Wang et al. (2016) have demonstrated that the summertime $O_3$ mass concentration holds a high level in 2014 in Beijing, with a daily

**54** average of up to 110 μg m$^{-3}$. Chen et al. (2015) have further shown that the average

**55** maximum daily $O_3$ concentrations are higher than 150 μg m$^{-3}$ during the summer in 2015 at

**56** most of monitoring sites in Beijing.

**57**     In recent years, Beijing has implemented aggressive emission control strategies to

**58** ameliorate the air quality (Parrish and Zhu, 2009). Both $NO_x$ (NO+$NO_2$) and total VOCs

**59** (volatile organic compounds) in Beijing have decreased linearly since 2002, while the

**60** daytime average $O_3$ concentration still increases rapidly (Tang et al., 2009; Wang et al., 2012;

**61** Zhang et al., 2014). Zhang et al. (2014) have highlighted the importance of the

**62** trans-boundary transport and the cooperation with neighboring provinces to control the $O_3$

**63** level in Beijing. Pollutants transported from outside of Beijing and formed locally together

**64** determine the air quality in Beijing (Meng et al., 2006; Zhang et al., 2012).

**65**     Several studies have been performed to investigate the role of trans-boundary transport

**66** in the air quality of Beijing based on observational analyses and model simulations. Using the

**67** US EPA's Model-3/CMAQ model simulation in the Beijing area, Streets et al. (2007) have

**68** pointed out that Hebei Province can contribute 50-70% of Beijing's $PM_{2.5}$ concentration and

**69** 20-30% of $O_3$ concentration. Wang et al. (2009) have indicated that $O_3$ formation in Beijing

**70** is not only affected by local emissions, but also influenced by Tianjin and the south of Hebei

**71** Province. The intense regional transport of pollutants from south to north in NCP has been

**72** proposed to be the main reason for the heavy haze pollution in January 2013 in Beijing (Sun

**73** et al., 2014; Tao et al., 2014; Wang et al., 2014b). Jiang et al. (2015) have demonstrated that

**74** the transport from the environs of Beijing contributes about 55% of the peak $PM_{2.5}$

**75** concentration in the city during a heavy haze event in December 2013.

Since September 2013, the 'Atmospheric Pollution Prevention and Control Action Plan'
(hereafter referred to as APPCAP) has been implemented, which is released by the Chinese
State Council to reduce $PM_{2.5}$ by up to 25% by 2017 relative to 2012 levels. After
implementation of the APPCAP, high $PM_{2.5}$ mass concentrations still can be observed and
the $O_3$ pollution has deteriorated during summertime since 2013 in Beijing (Chen et al., 2015;
Wang et al., 2016). Hence, studies are imperative to explore the $O_3$ and $PM_{2.5}$ formation from
various sources and evaluate the pollutants contributions from local production and
trans-boundary transport in Beijing, to support the design of mitigation strategies.
The purpose of the present study is to evaluate the contributions of trans-boundary
transport of emissions outside of Beijing to the air quality in Beijing and interaction of
emissions in and outside of Beijing after APPCAP using the WRF-CHEM model. The model
configuration and methodology are described in Section 2. Model results and sensitivity
studies are presented in Section 3, and conclusions and discussions are given in Section 4.

**2    Model and Methodology**
**2.1   WRF-CHEM Model**
The WRF-CHEM model used in the study is developed by Li et al. (2010, 2011a, b,
2012) at the Molina Center for Energy and the Environment, with a new flexible gas phase
chemical module and the CMAQ aerosol module developed by US EPA. The aerosol
component of the Community Multiscale Air Quality (CMAQ) model is designed to be an
efficient and economical depiction of aerosol dynamics in the atmosphere (Binkowski and
Roselle, 2003). The particle size distribution in the study is represented as the superposition

**98**  of three lognormal subdistributions, called modes, which includes the processes of

**99**  coagulation, particle growth by the addition of mass, and new particle formation. Following

**100**  the work of Kulmala et al. (1998), the new particle production rate presented here is

**101**  calculated as a parameterized function of temperature, relative humidity, and the vapor-phase

**102**  $H_2SO_4$ concentration due to binary nucleation of $H_2SO_4$ and $H_2O$ vapor, and the new particles

**103**  are assumed to be 2.0 nm diameter. A number of recent studies have shown that organic

**104**  compounds can play an important role in nucleation process (Zhang et al., 2009, 2012, 2015).

**105**  The contribution from organic acids likely explains the high levels of aerosol, especially in

**106**  polluted urban area, where large amount of organic acids can be emitted directly and

**107**  produced by photochemical oxidation of hydrocarbons (Fan et al., 2006), which needs to be

**108**  considered in the further study. The wet deposition follows the method used in the CMAQ

**109**  and the surface deposition of chemical species is parameterized following Wesely (1989).

**110**  The photolysis rates are calculated using the FTUV (Li et al., 2005; Li et al., 2011a), in

**111**  which the effects of aerosols and clouds on photolysis are considered.

**112**  The inorganic aerosols are predicted in the WRF-CHEM model using ISORROPIA

**113**  Version 1.7 (Nenes et al., 1998). The efficient and rapid secondary species formation in

**114**  Beijing has been found during the severe haze formation process in the previous study (Guo

**115**  et al., 2014). The secondary organic aerosol (SOA) formation is calculated using a

**116**  non-traditional SOA module. The volatility basis-set (VBS) modeling method is used in the

**117**  module, assuming that primary organic components are semi-volatile and photochemically

**118**  reactive and are distributed in logarithmically spaced volatility bins. Detailed information

**119**  about the volatility basis-set approach can be found in Li et al (2011b). Recent studies have

**120** shown that small di-carbonyls (glyoxal and methylglyoxal) are important for the aerosol

**121** formation due to their traffic origin (Zhao et al., 2006; Gomez et al., 2015). Li et al. (2011a)

**122** have indicated that glyoxal and methylglyoxal can contribute about 10% of the SOA in the

**123** urban area of Mexico City. The SOA formation from glyoxal and methylglyoxal in this study

**124** is parameterized as a first-order irreversible uptake by aerosol particles and cloud droplets,

**125** with a reactive uptake coefficient of $3.7 \times .7^{-3}$ for glyoxal and methylglyoxal (Zhao et al.,

**126** 2006; Volkamer et al., 2007; Gomez et al., 2015).

**127** ==**2.2    Pollution Episode Simulation**==

**128** A persistent air pollution episode from 5 to 14 July 2015 in Beijing-Tianjin-Hebei

**129** (BTH) is simulated using the WRF-CHEM model. During the episode, the observed mean

**130** daily $PM_{2.5}$ concentration is 73.8 $\mu g \ m^{-3}$ and the average $O_3$ concentration in the afternoon

**131** reaches 237.0 $\mu g \ m^{-3}$ in Beijing. The maximum of $O_3$ concentration is higher than 350 $\mu g \ m^{-3}$,

**132** and the maximum of $PM_{2.5}$ concentration can reach a high level exceeding 150 $\mu g \ m^{-3}$.

**133** SI-Figures 1a-c show the daily averages of the temperature, relative humidity, and wind

**134** speed in Beijing during the summer of 2015. The minimum air temperature is 18.7°C, and the

**135** maximum air temperature is 40 °C during the summer, with average of 25.7°C. The average

**136** relative humidity is 63.8%. The southeast or southwest wind is prevailing over NCP due to

**137** the influence of East Asian summer monsoon (Zhang et al., 2010), with the average wind

**138** speed of 5.6 $m \ s^{-1}$ in the summer of 2015. During the study period, the average temperature,

**139** relative humidity, and wind speed are 28.4°C, 51.7% and 6.3 $m \ s^{-1}$, respectively, indicating

**140** typical summertime meteorological conditions. During the summer of 2015, the average

**141** $PM_{2.5}$ concentration is 56.1 $\mu g \ m^{-3}$ and the average $O_3$ concentration in the afternoon is 216.4

**142** μg m$^{-3}$ (SI-Figures 1d-e). The high O$_3$ and PM$_{2.5}$ event occurs frequently during the

**143** summertime of 2015, so the study period can well represent the summertime O$_3$ and PM$_{2.5}$

**144** pollution in Beijing, and provide a suitable case for observation analyses and model

**145** simulations to investigate the effect of trans-boundary transport on the summertime air

**146** quality of Beijing.

**147** The WRF-CHEM model adopts one grid with horizontal resolution of 6 km and 35

**148** sigma levels in the vertical direction, and the grid cells used for the domain are 200 × 200

**149** (Figure 1). The physical parameterizations include the microphysics scheme of Hong et al

**150** (Hong and Lim, 2006), the Mellor, Yamada, and Janjic (MYJ) turbulent kinetic energy (TKE)

**151** planetary boundary layer scheme (Janjić, 2002), the Unified Noah land-surface model (Chen

**152** and Dudhia, 2001), the rapid radiative transfer model (RRTM) long wave radiation scheme

**153** (Mlawer et al., 1997) and the Goddard shortwave parameterization (Suarex and Chou, 1994;

**154** Chou and Suarez, 1999, 2001). The NCEP 1° × 1° reanalysis data are used to obtain the

**155** meteorological initial and boundary conditions, and the meteorological simulations are not

**156** nudged in the study. The chemical initial and boundary conditions are interpolated from the

**157** 6h output of MOZART (Horowitz et al., 2003). The spin-up time of the WRF-CHEM model

**158** is 28 hours. The SAPRC-99 (Statewide Air Pollution Research Center, version 1999)

**159** chemical mechanism is used in the present study.

**160** The anthropogenic emissions are developed by Zhang et al. (2009), which is based on

**161** the 2013 emission inventory, including contributions from agriculture, industry, power

**162** generation, residential, and transportation sources. The SO$_2$, NO$_x$, and CO emissions have

**163** been adjusted according to their observed trends from 2013 to 2015 in the present study, but

the VOCs emissions are not changed considering that the VOCs emissions are still not fully
considered in the current air pollutant control strategy. The major pollutants emissions used
in the model simulation for Beijing, Tianjin, and the neighboring provinces (Hebei, Shanxi,
and Shandong) are summarized in Table 1. Obviously, high anthropogenic emissions are
distributed outside of Beijing, especially in Hebei and Shandong provinces. Figure 2 presents
distributions of the emission rates of VOCs, $NO_x$, OC, and $SO_2$ in the simulation domain,
showing that the anthropogenic emissions are generally concentrated in urban areas. As
shown in Figure 2, the total emissions from neighboring regions are much more than those in
Beijing, and the emission rates in Tianjin, the south of Hebei and Shandong are also higher
than those in Beijing, particularly with regard to $SO_2$ emissions. Therefore, when the south or
east wind is prevailing in NCP, the severe air pollution can be formed in Beijing when
precursor emissions in highly industrialized areas chemically react as they are carried toward
Beijing, blocked by mountains and further accumulated and interacted with those in Beijing.
It is worth noting that uncertainties of the emission inventory used in the study are still rather
large taking consideration of the rapid changes in anthropogenic emissions that are not fully
reflected in the current emission inventories, particularly since implementation of the
APPCAP, and the complexity of pollutants precursors. For example, different VOCs types
exhibit distinct kinetic behaviors, and as an important fraction of total VOCs in the urban
atmosphere, aromatics are responsible for the photochemical ozone production and secondary
organic aerosol formation (Suh et al., 2003; Fan et al., 2004). In the SAPRC99, aromatics are
lumped into ARO1 and ARO2. ARO1 mainly includes toluene, benzene, ethylbenzene, and
other aromatics with reaction rate with OH (kOH) less than $2\times10^4$ $ppm^{-1}$ $min^{-1}$. ARO2

**186** includes xylene, trimethylbenzene, and other aromatics with kOH greater than $2\times10^4$ ppm$^{-1}$

**187** min$^{-1}$. Additionally, biogenic VOCs also play a considerable role in the ozone production (Li

**188** et al., 2007), and monoterpenes and isoprene are the main biogenic VOCs in the SAPRC99

**189** chemical mechanism. The biogenic emissions are calculated online using the MEGAN

**190** (Model of Emissions of Gases and Aerosol from Nature) model developed by Guenther et al

**191** (2006).

**192** ### 2.3   Factor Separation Approach

**193**    The formation of the secondary atmospheric pollutant, such as O$_3$, secondary organic

**194** aerosol, and nitrate, is a complicated nonlinear process in which its precursors from various

**195** emission sources and transport react chemically or reach equilibrium thermodynamically.

**196** Nevertheless, it is not straightforward to evaluate the contributions from different factors in a

**197** nonlinear process. The factor separation approach (FSA) proposed by Stein and Alpert (1993)

**198** can be used to isolate the effect of one single factor from a nonlinear process and has been

**199** widely used to evaluate source effects (Gabusi et al., 2008; Weinroth et al., 2008; Carnevale

**200** et al., 2010; Li et al., 2014). The total effect of one factor in the presence of others can be

**201** decomposed into contributions from the factor and that from the interactions of all those

**202** factors.

**203**    Suppose that field $f$ depends on a factor $\varphi$:

**204**    $$f = f(\varphi)$$

**205** The FSA decomposes function $f(\varphi)$ into a constant part that does not depend on $\varphi$ ($f(0)$)

**206** and a $\varphi$-depending component ($f'(\varphi)$), as follows:

**207**    $$f'(0) = f(0)$$

$f'(\varphi) = f(\varphi) - f(0)$
Considering that there are two factors $X$ and $Y$ that influence the formation of secondary
pollutants in the atmosphere and also interact with each other. Denoting $f_{XY}$, $f_X$, $f_Y$, and
$f_0$ as the simulations including both of two factors, factor $X$ only, factor $Y$ only, and none of
the two factors, respectively. The contributions of factor $X$ and $Y$ can be isolated as follows:
$f'_X = f_X - f_0$
$f'_Y = f_Y - f_0$
Note that term $f'_{X(Y)}$ represents the impacts of factor $X(Y)$, while $f_0$ is the term
independent of factors $X$ and $Y$.
The simulation including both factors $X$ and $Y$ is given by:
$f_{XY} = f_0 + f'_X + f'_Y + f'_{XY}$
The mutual interaction between $X$ and $Y$ can be expressed as:
$f'_{XY} = f_{XY} - f_0 - f'_X - f'_Y = f_{XY} - (f_X - f_0) - (f_Y - f_0) - f_0 = f_{XY} - f_X - f_Y + f_0$
The above equation shows that the study needs four simulations, $f_{XY}, f_X, f_Y$ and $f_0$, to
evaluate the contributions of two factors and their synergistic interactions.
**2.4 Statistical Metrics for Observation-Model Comparisons**
In the present study, the mean bias ($MB$), root mean square error ($RMSE$) and the index
of agreement ($IOA$) are used as indicators to evaluate the performance of WRF-CEHM model
in simulation against measurements. $IOA$ describes the relative difference between the model
and observation, ranging from 0 to 1, with 1 indicating perfect agreement.
$MB = \frac{1}{N}\sum_{i=1}^{N}(P_i - O_i)$
$RMSE = \left[\frac{1}{N}\sum_{i=1}^{N}(P_i - O_i)^2\right]^{\frac{1}{2}}$
$$IOA = 1 - \frac{\sum_{i=1}^{N}(P_i - O_i)^2}{\sum_{i=1}^{N}(|P_i - \overline{O}| + |O_i - \overline{O}|)^2}$$
where $P_i$ and $O_i$ are the predicted and observed pollutant concentrations, respectively. $N$ is
the total number of the predictions used for comparisons, and $\overline{P}$ and $\overline{O}$ represents the
average of the prediction and observation, respectively.
**2.5 Pollutant Measurements**
The hourly measurements of $O_3$, $NO_2$, and $PM_{2.5}$ used in the study are downloaded
from the website http://www.aqistudy.cn/. The submicron sulfate, nitrate, ammonium, and
organic aerosols are observed by the Aerodyne Aerosol Chemical Speciation Monitor
(ACSM), which is deployed at the National Center for Nanoscience and Technology
(NCNST), Chinese Academy of Sciences, Beijing (Figure 1). The mass spectra of organic
aerosols are analyzed using the Positive Matrix Factorization (PMF) technique to separate
into four components: hydrocarbon-like organic aerosol (HOA),cooking organic aerosol
(COA),coal combustion organic aerosol (CCOA), and oxygenated organic aerosol (OOA).
HOA, COA, and CCOA are interpreted as surrogates of primary organic aerosol (POA), and
OOA is a surrogate of SOA.
The APPCAP has been implemented since 2013 September, so comparisons of
summertime pollutants between 2013 and 2015 can show the mitigation effects on the air
quality. Considering that high $O_3$ concentrations generally take place in the afternoon during
summertime, Table 2 presents the summertime concentrations of pollutants in the afternoon
(12:00 – 18:00 Beijing Time (BJT)) averaged at 12 monitoring sites in Beijing in 2013 and
2015. The rainy days during summertime in Beijing are 43 and 46 days in 2013 and 2015,
respectively, showing the similar meteorological conditions between the two years. Therefore,
in general, the air pollutants variations between 2013 and 2015 can be mainly attributed to
implementation of the APPCAP. Apparently, implementation of the APPCAP has
considerably decrease the concentrations of primary species of CO and $SO_2$, particularly with
regard to $SO_2$, reduced by more than 40% from 2013 to 2015. Most of $NO_x$ exist in the form
of $NO_2$ in the afternoon during summertime due to active photochemical processes. Therefore,
25.1% decrease of $NO_2$ in the afternoon from 2013 to 2015 shows that the $NO_x$ emission
mitigation is also effective in Beijing. The $PM_{2.5}$ concentrations are decreased by about 24.0%
from 2013 to 2015, approaching the expected 25% reduction by 2017 relative to 2012 levels.
However, the $O_3$ trend is not anticipated in Beijing, and $O_3$ concentrations are increased from
133.0 $\mu g\ m^{-3}$ in 2013 to 163.2 $\mu g\ m^{-3}$ in 2015, enhanced by 22.8%. For the discussion
convenience, we have defined the $O_3$ exceedance with hourly $O_3$ concentrations exceeding
200 $\mu g\ m^{-3}$ and $PM_{2.5}$ exceedance with hourly $PM_{2.5}$ concentrations exceeding 75 $\mu g\ m^{-3}$.
Although the $PM_{2.5}$ exceedance frequency in the afternoon has been decreased by 25.0%
from 2013 to 2015, but still remains 32.7% in 2015. The $O_3$ exceedance frequency in 2015 is
31.8%, enhanced by 57.6% compared to 20.2% in 2013. Hence, during the summertime of
2015, two years after implementation of the APPCAP, Beijing still has experienced high $O_3$
and/or $PM_{2.5}$ pollutions frequently.

**3      Results and Discussions**
**3.1    Model Performance**
The hourly measurements of $O_3$, $NO_2$, and $PM_{2.5}$ in Beijing-Tianjin-Hebei (BTH) and
ACSM measured aerosol species in Beijing are used to validate the WRF-CHEM model
simulations.

### 3.1.1 $O_3$, $NO_2$, and $PM_{2.5}$ Simulations in Beijing

Figure 3 shows the temporal variations of observed and simulated near-surface $O_3$, $NO_2$,
and $PM_{2.5}$ concentrations averaged over monitoring sites in Beijing from 5 to 14 July 2015.
The WRF-CHEM model performs reasonably well in simulating the $PM_{2.5}$ variations
compared with observations in Beijing. The *MB* and *RMSE* are -3.6 μg m$^{-3}$ and 22.5 μg m$^{-3}$,
respectively, and the *IOA* is 0.86. The model well reproduces the temporal variations of $O_3$
concentrations, with an *IOA* of 0.92. The model considerably underestimates the $O_3$
concentration during daytime on July 5, 6 and 13. Most of monitoring sites in Beijing are
concentrated in the urban area. Therefore, if the simulated winds cause the $O_3$ plume formed
in the urban area to leave early or deviate the $O_3$ plume transported from outside of Beijing
from the urban area, the model is subject to underestimate the $O_3$ concentration in Beijing
(Bei et al., 2010). The WRF-CHEM model also reasonably yields the $NO_2$ diurnal profiles,
but frequently overestimates the $NO_2$ concentrations during nighttime, which is likely caused
by the biased boundary layer simulations.

### 3.1.2 Aerosol Species Simulations in Beijing

Figure 4 shows the temporal variations of simulated and observed aerosol species at
NCNST site in Beijing from 5 to 14 July 2015. The WRF-CHEM model generally performs
reasonably in simulating the aerosol species variations compared with ACSM measurements.
As a primary aerosol species, the POA in Beijing is determined by direct emissions from
various sources and transport from outside of Beijing, so uncertainties from emissions and
meteorological fields remarkably affect the model simulations (Bei et al., 2012; Bei et al.,

**296**  2013). Although the *MB* and *RMSE* for POA are 0.0 μg m$^{-3}$ and 3.1 μg m$^{-3}$, respectively, the

**297**  *IOA* is less than 0.60, indicating the considerable biases in POA simulations. The

**298**  WRF-CHEM model has difficulties in well simulating the sulfate aerosol, with an *IOA* lower

**299**  than 0.60. The model cannot produce the observed high peaks of sulfate aerosols around

**300**  noontime on 8, 11, and 12 July 2015. The sulfate aerosol in the atmosphere is produced from

**301**  multiple sources, including SO$_2$ gas-phase oxidations by hydroxyl radicals (OH) and

**302**  stabilized criegee intermediates (sCI), aqueous reactions in cloud or fog droplets, and

**303**  heterogeneous reactions on aerosol surfaces, as well as direct emissions from power plants

**304**  and industries (Li et al., 2016). The model reasonably well reproduces the observed temporal

**305**  variations of SOA, nitrate, and ammonium, with *IOA*s exceeding 0.75. The model simulate

**306**  well the peak concentration of SOA, nitrate and ammonium at the rush hour, but the model

**307**  also underestimates the SOA, nitrate and ammonium as well, with *MB* of -1.1 μg m$^{-3}$, -0.7 μg

**308**  m$^{-3}$, and -0.5 μg m$^{-3}$, respectively. For nitrate and ammonium, the underestimates occur

**309**  mainly on 8 July 2015 possibly due to wind filed, which will be further analyzed in

**310**  supplement (SI-Figure 2).

**311**  **3.1.3 O$_3$, NO$_2$, and PM$_{2.5}$ Simulations in BTH**

**312**  Figure 5 shows the diurnal profiles of observed and simulated near-surface O$_3$, NO$_2$,

**313**  and PM$_{2.5}$ concentrations averaged over monitoring sites in BTH from 5 to 14 July 2015. The

**314**  WRF-CHEM model exhibits good performance in predicting the temporal variations of O$_3$,

**315**  NO$_2$, and PM$_{2.5}$ concentrations compared with measurements in BTH, with *IOA*s higher than

**316**  0.80. In addition, O$_3$ and NO$_2$ simulations are also improved in BTH compared to those in

**317**  Beijing, indicating better model performance for regional simulations in a large scale.

Figure 6 presents the distributions of calculated and observed near-surface $PM_{2.5}$
concentrations along with the simulated wind fields at 10:00 Beijing Time (BJT) on the six
selected representative days with high $O_3$ and $PM_{2.5}$ concentrations. The calculated $PM_{2.5}$
spatial patterns generally agree well with the observations at the monitoring sites. The
observed $PM_{2.5}$ concentrations in BTH are still high even after implementation of the
APPCAP, frequently exceeding 75 μg $m^{-3}$ on the selected six days. The $PM_{2.5}$ concentrations
in Beijing are higher than 115 μg $m^{-3}$ at 10:00 BJT on 8, 11, and 12 July 2015, causing
moderate air pollution.
The $O_3$ concentration during summertime reaches its peak during the period from 14:00
to 16:00 BJT in Beijing (Tang et al., 2012). Figure 7 presents the spatial distribution of
calculated and measured near-surface $O_3$ concentration at 15:00 Beijing Time (BJT) on the
selected six days, along with the simulated wind fields. In general, the simulated $O_3$ spatial
patterns are consistent with the measurements, but model biases still exist. High $O_3$
concentrations at 15:00 BJT in Beijing are observed and also simulated by the model,
frequently exceeding 250 μg $m^{-3}$. The $O_3$ transport to Beijing from its surrounding areas is
also obvious when the winds are easterly or southerly. Figure 8 provides the spatial
distribution of simulated and observed near-surface $NO_2$ concentration on the selected six
days at 08:00 BJT when the $NO_2$ concentration reaches it peak due to rush hour $NO_x$
emissions and low planetary boundary layer (PBL). The simulated near-surface $NO_2$
concentrations highlights the dominant impact of the anthropogenic emissions, primarily
concentrated in cities or their downwind areas, which generally agree well with the
measurements. Beijing is surrounded from south to east by cities with high $NO_2$
concentrations, which can influence the $O_3$ formation in Beijing when south or east winds are
prevalent.
The good agreements between predicted $PM_{2.5}$, $O_3$, $NO_x$ and aerosol species and the
corresponding measurements show that the modeled meteorological fields and emissions
used in simulations are generally reasonable.
**3.2    Contributions of Trans-boundary Transport to the $O_3$ and $PM_{2.5}$ Levels in Beijing**
**3.2.1 Analysis of horizontal transport of $O_3$ and $PM_{2.5}$**
The analysis in Section 3.1.3 has shown the strong correlation between the airflow and
the high level of pollutants in Beijing during the study episode. It is essential to confirm
whether the continuous air pollutions in Beijing are directly related to the airflow transport
from outside of Beijing (An et al., 2007; Yang et al., 2010). In the present study, the
horizontal transport flux intensity is defined as the horizontal wind speed on the grid border
multiplied by the pollutants concentration of the corresponding grid from which the airflows
comes (Jiang et al., 2008). Considering that trans-boundary transport mainly occurs within
the PBL, the study also focuses on the contribution of trans-boundary transport of pollutants
within PBL over Beijing and its surrounding areas. Previous studies have shown that the
average mixing layer height is approximately between 600—800 m during summertime, with
the maximum during noontime higher than 1000 m (Wang et al.,2015; Tang et al., 2016).
Figure 9 shows the temporal variations of net horizontal transport flux of $PM_{2.5}$, $O_3$ and $NO_2$
through Beijing boundary and the pollutants contributions from non-Beijing emissions to the
air quality in Beijing city. The hourly $PM_{2.5}$, $O_3$ and $NO_2$ contributions of non-Beijing
emissions generally have the same variation trend as the horizontal transport flux, indicating
that the contribution of surrounding sources plays an important role in high pollutants
concentrations in Beijing during the study episode. For example, the $O_3$ net flux also has the
similar peak in the afternoon as the $O_3$ contribution from the non-Beijing emissions. As
discussed in Section 3.1.3, the prevailing south wind dominates in BTH, so the largest flux
intensity are from the south, with the average of 103.3 g s$^{-1}$ and 244.5 g s$^{-1}$ for $PM_{2.5}$ and $O_3$,
respectively (SI-Table 1), indicating that the pollutants are mainly from the south. It should
be noted that the flux of $O_3$ is mainly focused on the afternoon from 12:00 to 18:00 BJT. The
average net horizontal transport fluxes for $PM_{2.5}$ and $O_3$ during the episode are 68.2 g s$^{-1}$ and
68.5 g s$^{-1}$, respectively, showing important contributions of non-Beijing emissions to the air
quality in Beijing.

**3.2.2 Trans-boundary transport contributions to $O_3$ in Beijing**

The FSA is used in the present study to evaluate the contributions and interactions of
emissions from Beijing and outside of Beijing to the near-surface concentrations of $O_3$ and
$PM_{2.5}$ in Beijing. Four model simulations are performed, including $f_{BS}$ with both the
anthropogenic emissions from Beijing and outside of Beijing, $f_B$ with the emission from
Beijing alone, $f_S$ with only emissions outside of Beijing, and $f_0$ without both the
emissions from Beijing and outside of Beijing, representing background concentrations.
Apparently, the air pollutants levels in Beijing are determined by the contribution from local
emissions ($f'_B$, $f_B - f_0$), the trans-boundary transport of non-Beijing emissions ($f'_S$,
$f_S - f_0$), emission interactions between local and non-Beijing emissions ($f'_{BS}$, $f_{BS} - f_B -$
$f_S + f_0$), and background ($f_0$).
Figure 10 provides the temporal variations of the average near-surface $O_3$ and $PM_{2.5}$
concentrations from $f_{BS}$ with all the emissions, $f_B$ with Beijing emissions alone, and $f_S$
with non-Beijing emissions alone in Beijing from 5 to 14 July 2015. Apparently, non-Beijing
emissions generally play a more important role in the $O_3$ level of Beijing than local emissions.
Even when the Beijing local emissions are excluded, the $O_3$ concentration in Beijing still
remains high level, with an average of 153 μg m$^{-3}$ and ranging from 130 to 180 μg m$^{-3}$ in the
afternoon. When only considering the Beijing local emission in simulations, the afternoon
average $O_3$ concentration in Beijing is approximately 126.6 μg m$^{-3}$, varying from 80 to 160
μg m$^{-3}$. On July 13, the contribution from Beijing local emissions exceeds that from
non-Beijing emissions because north winds are prevailing, bringing clean air to Beijing
(Figure 7f). Table 3 gives the average $O_3$ contributions from 12:00 to 18:00 BJT in Beijing
from local emissions, non-Beijing emissions, emission interactions, and background. The
local emissions contribute about 22.4% on average in the afternoon to the $O_3$ level in Beijing,
varying from 15.5% to 35.4%. The non-Beijing emissions contribute more than local sources,
with an average contribution of 36.6%, ranging from 15.2% to 48.0%. The emission
interactions in Beijing decrease the $O_3$ level by 5.1% on average. $O_3$ formation is a nonlinear
process, depending on not only the absolute levels of $NO_x$ and VOCs, but also the ratio of
$VOC_s/NO_x$ (Sillman et al., 1990; Lei et al., 2007, 2008). When the $O_3$ precursors emitted
from outside of Beijing are transported to Beijing and mixed with local emissions, the
concentrations of $O_3$ precursors are increased and the ratio of $VOC_s/NO_x$ is also altered,
causing the formed $O_3$ concentration unequal to the simple linear summation of $O_3$
contributions from the local and non-Beijing emissions. The background $O_3$ in Beijing plays
an important role in the $O_3$ level in the afternoon, accounting for 46.1% of the $O_3$
concentration. The background $O_3$ contribution varies from 32.6% to 62.9% during the
episode, which is primarily determined by the prevailing wind direction. When the northerly
wind is prevalent, the clean airflow from the north affects Beijing, enhancing the background
$O_3$ contribution, such as on 5, 13, and 14 July 2015. However, when the polluted airflow
from the south impacts Beijing, the background $O_3$ contribution is decreased. The $O_3$
contributions in Beijing induced by the trans-boundary transport of emissions outside of
Beijing is about 31.5% of the $O_3$ concentration during the study episodes, which is in
agreement with previous studies (Streets et al., 2007; Wang et al., 2008), indicating that the
trans-boundary transport constitutes the main reason for the elevated $O_3$ level in Beijing after
implementation of the APPCAP.
Previous studies have proposed that the regional transport of $O_3$ precursors can play an
important role in inducing the high $O_3$ concentrations level in Beijing (Wang et al., 2009;
Zhang et al., 2014). SI-Table 2 provides the average $NO_2$ contributions in Beijing from local
emissions, non-Beijing emissions, emission interactions, and background. Different from $O_3$,
the local emissions dominate the level of $NO_2$ in Beijing area, with an average contribution of
70.3% during the study episode. The average contribution of non-Beijing emissions, emission
interactions and background are 24.8%, 0.9% and 4.0%, respectively. Compared to the direct
input of regional $O_3$, the regional transport of $NO_x$ is unlikely a significant contributor to high
$O_3$ concentrations in Beijing, partly due to its short lifetime in the summer.
**3.2.3 Trans-boundary transport contributions to $PM_{2.5}$ in Beijing**
When the Beijing local emissions are not considered in simulations, Beijing still
experiences high $PM_{2.5}$ pollution, with an average $PM_{2.5}$ concentration of 48.3 μg m$^{-3}$ during
the simulation episode, and the PM$_{2.5}$ level in Beijing still exceeds 75 μg m$^{-3}$ on several days.
However, when only considering the Beijing local emissions, the average PM$_{2.5}$
concentration in Beijing is 19.6 μg m$^{-3}$ during the episode, showing that Beijing's PM$_{2.5}$
pollution is dominated by the trans-boundary transport (Figure 10b). Table 4 shows the
average PM$_{2.5}$ contribution in Beijing from local emissions, non-Beijing emissions, emission
interactions, and background. During the study episode, the average PM$_{2.5}$ contribution from
local emissions is 13.7%, which is much lower than the contribution of 61.5% from
non-Beijing emissions, further showing the dominant role of the trans-boundary transport in
the Beijing PM$_{2.5}$ pollution. The emission interactions enhance the PM$_{2.5}$ level in Beijing on
average, with a contribution of 5.9%. The background PM$_{2.5}$ contribution to Beijing is 18.9%
on average, lower than those for O$_3$. The PM$_{2.5}$ contribution caused by the trans-boundary
transport is about 67.4% of PM$_{2.5}$ concentrations in Beijing, indicating that the cooperation
with neighboring provinces to control the PM$_{2.5}$ level is a key for Beijing to improve air
quality. Previous studies have also demonstrated the dominant role of non-Beijing emissions
in the PM$_{2.5}$ level in Beijing. Based on CMAQ model, Streets et al., (2007) have reported that
average contribution of regional transport to PM$_{2.5}$ at the Olympic Stadium can be 34%, up to
50%—70% under prevailing south winds. Guo et al. (2010) have provided a rough estimation
that the regional transport can contribute 69% of the PM$_{10}$ and 87% of the PM$_{1.8}$ in Beijing
local area using the short and low time resolution data in the summer. Combining the PM$_{2.5}$
observations and MM5-CMAQ model results, regional transport is estimated to contribute
54.6% of the PM$_{2.5}$ concentration during the polluted period, with an annual average PM$_{2.5}$
contribution of 42.4% (Lang et al., 2013). Using the long-term measurements of PM$_{2.5}$ mass
concentrations from 2005 to 2010 at urban Beijing, and trajectory cluster and receptor models,
the average contribution of long-distance transport to Beijing's PM$_{2.5}$ level can be
approximately 75.2% in the summer (Wang et al., 2015).

### 3.2.4 Trans-boundary transport contributions to aerosol species in Beijing

Figure 11 shows the temporal variation of the averaged contributions to the near-surface
aerosol constituents from total emissions ($f_{BS}$), local emissions ($f'_B$), the trans-boundary
transport of non-Beijing emissions ($f'_S$), emission interactions ($f'_{BS}$), and the background
($f_0$) during the simulation episode. The temporal variations of elemental carbon (EC) and
POA from local emissions and trans-boundary transport exhibit obvious diurnal cycles, i.e.,
highest during nighttime and lowest in the afternoon, corresponding to the variations of PBL
height and anthropogenic emissions. The SOA from local emissions reaches its peak in the
afternoon when the O$_3$ concentration is high, but the trans-boundary transport causes the
gradual accumulation process of SOA in Beijing from July 5 to 9 and from July 9 to 13. The
sulfate temporal profile from the trans-boundary transport is similar to that of SOA, also
showing the accumulation process. In addition, the sulfate aerosols from local emissions do
not vary remarkably. The nitrate aerosols from local emissions and the trans-boundary
transport generally attain peaks in the morning when the air temperature is not high and the
HNO$_3$ concentrations are not low. The ammonium aerosol variations are generally
determined by those of sulfate and nitrate aerosols. For example, the variations of ammonium
aerosols from the trans-boundary include not only the morning peaks, but also the
accumulation processes from July 5 to 9 and from July 9 to 13. Except the sulfate aerosol, the
temporal variations of aerosol species from background are not large.

472       Table 5 presents the average aerosol constituents contributions from Beijing local

emissions, non-Beijing emissions, emission interactions, and the background, and mass
fractions in the total $PM_{2.5}$ in Beijing during the episode. Organic aerosols (POA+SOA)
constitute the most important component of $PM_{2.5}$, accounting for 34.8% of $PM_{2.5}$ mass
concentration, which is consistent with the ACSM measurement in Beijing (Sun et al., 2014).
In addition, SOA contributes more than 70% of organic aerosol mass concentrations, which is
due to the increased atmospheric oxidation capability caused by elevated $O_3$ concentrations
during summertime. Although the $SO_2$ concentrations have been decreased by more than 40%
since implementation of the APPCAP, sulfate aerosols still play an important role in the
$PM_{2.5}$ level in Beijing and make up 25.1% of the $PM_{2.5}$ mass concentrations, showing high
sulfate contributions from the trans-boundary transport and background. The ammonium,
nitrate, EC, and unspecified species account for 13.7%, 14.1%, 5.8%, and 6.5% of the $PM_{2.5}$
mass concentrations, respectively. Secondary aerosol species dominate the $PM_{2.5}$ mass
concentration in Beijing, with a contribution of 77.9%.

486       The local emissions contribute more than 20% of the mass concentrations for the

primary aerosol species, but less than 15% for the secondary aerosol species in Beijing (Table
5). The trans-boundary transport of non-Beijing emissions dominates all the aerosol species
levels in Beijing, with contributions exceeding 50%, particularly for SOA and nitrate. In
addition, the POA and sulfate background contributions are also high, more than 20%.
Although the primary aerosol species of EC and unspecified constituents are not involved in
the chemical process and also do not participate in the gas-particle partitioning, the emission
interactions still enhance EC and unspecified constituents concentrations, with contributions

of around 1.5%, which is caused by the PBL-pollution interaction. It is clear that the PBL-pollution interaction plays an important role in the pollutant accumulation in Beijing (Wang et al., 2013; Peng et al., 2016). Mixing of Beijing local emissions with those outside of Beijing increases the aerosol concentrations in the PBL and decreases the incoming solar radiation down to the surface, cooling the temperature of the low level atmosphere to suppress the development of PBL and hinder the aerosol dispersion in the vertical direction.

The emission interactions increase the POA and SOA concentrations, with a POA contribution of 5.3% and a SOA contribution of 5.9%. In the VBS modeling approach, primary organic components are assumed to be semi-volatile and photochemically reactive. Mixing of Beijing local emissions with non-Beijing emissions enhances the organic condensable gases, and considering that the saturation concentrations of the organic condensable gases do not change, more organic condensable gases participate into the particle phase, increasing the POA and SOA concentrations.

The contributions of emission interactions to inorganic aerosols, including sulfate, nitrate, and ammonium are more complicated, depending on their particle phase and precursors concentrations. In the present study, ISORROPIA (Version 1.7) is used to calculate the thermodynamic equilibrium between the sulfate-nitrate-ammonium-water aerosols and their gas phase precursors $H_2SO_4$-$HNO_3$-$NH_3$-water vapor. Although mixing of Beijing local emissions with non-Beijing emissions increases inorganic aerosols precursors, the inorganic aerosol contributions from emission interactions are still uncertain due to the deliberate thermodynamic equilibrium between inorganic aerosols and their precursors. High atmospheric oxidation capability induced by elevated $O_3$ concentration facilitates $HNO_3$

formation through $NO_2$ reaction with OH during daytime and $N_2O_5$ formation through $NO_2$ reaction with $O_3$ during nighttime. High $O_3$ concentrations are produced by Beijing local emissions and non-Beijing emissions, accelerating the $HNO_3$ or $N_2O_5$ formation. Hence, mixing of Beijing local emissions with non-Beijing emissions considerably increases the $HNO_3$ or $N_2O_5$ levels, pushing more $HNO_3$ or $N_2O_5$ into the particle phase. The nitrate contributions from emission interactions are 18.1%, much more than those for other aerosol constituents. $SO_2$ gas-phase oxidations by OH and sCI are not as fast as $NO_2$ reaction with OH, so the formation of sulfuric acid is slow, although the $O_3$ concentration is high during summertime. Therefore, the sulfate contributions from emission interactions is not significant, only 3.4%. As the ammonium precursor, $NH_3$ is generally from direct emissions. The ammonium contributions from emission interactions are 1.5%, similar to those of primary aerosol species that are caused by PBL-pollution interactions, indicating that the $NH_3$ emissions are not sufficiently high in Beijing and outside of Beijing.

## 4    Summary and Conclusions

In the present study, a persistent air pollution episode with high concentrations of $O_3$ and $PM_{2.5}$ are simulated using the WRF-CHEM model during the period from July 5 to 14, 2015 in BTH, to evaluate the contributions of trans-boundary transport to the air quality in Beijing. Although the APPCAP has been implemented since 2013 September, the average $O_3$ concentration in the afternoon has been increased by 22.8% from 2013 to 2015 in Beijing, and Beijing still has experienced high $O_3$ and/or $PM_{2.5}$ pollutions frequently during summertime of 2015.

In general, the predicted temporal variations of $PM_{2.5}$, $O_3$, and $NO_2$ concentrations agree
well with observations in Beijing and BTH, but the model biases still exist, which is perhaps
caused by the uncertainties of simulated meteorological conditions and the emission
inventory. The model also successfully reproduces the spatial distributions of $PM_{2.5}$, $O_3$, and
$NO_2$ concentrations compared with measurements. The model performs reasonably well in
modeling the variations of aerosol constituents compared with ACSM measurement at
NCNST site in Beijing, but there are considerable biases in POA and sulfate simulations.
The FSA is used to investigate the contribution of trans-boundary transport of
non-Beijing emissions to the air quality in Beijing. If the Beijing local emissions are not
considered in model simulations, the $O_3$ and $PM_{2.5}$ concentrations in Beijing still remain high
levels, showing that the trans-boundary transport of emissions outside of Beijing plays a
more important role in the air quality in Beijing than the Beijing local emissions. On average,
the local emissions contribute 22.4% of $O_3$ in the afternoon and 13.7% of $PM_{2.5}$ mass
concentrations in Beijing during the episode. The $O_3$ contribution in the afternoon and $PM_{2.5}$
contribution from the trans-boundary transport of non-Beijing emissions are 36.6% and
61.5%, respectively, far exceeding those from local emissions. The interactions between local
and non-Beijing emissions generally decrease the $O_3$ level in the afternoon and increase the
$PM_{2.5}$ level in Beijing during the episode, with contributions of -5.1% and +4.4%,
respectively. In addition, the trans-boundary transport dominates all the aerosol species levels
in Beijing, with contributions exceeding 50% on average, particularly for SOA and nitrate.
The emission interactions in general increase all the aerosol species levels due to the
PBL-pollution interaction and the enhancement of precursors of secondary aerosols. Hence,
the air quality in Beijing during summertime is generally determined by the trans-boundary
transport of emissions outside of Beijing.
However, it is still controversial on whether local or non-local emissions play a
dominant role in the air quality in Beijing (Guo et al., 2010, 2014; Li et al., 2015; Zhang et al.,
2015). When only considering the local emissions, Beijing only experiences $O_3$ pollution, and
the $PM_{2.5}$ level is low during summertime, which is comparable to the air quality in Mexico
City. Mexico City has once been one of the most polluted cities in the world, but the air
quality has been greatly improved in recent years after taking emission control strategies
(Molina et al., 2002, 2007, 2010). Beijing and Mexico City now have similar emission
sources, including transportation and residential living, but Beijing is surrounded by the
highly industrialized areas in the south and east. When considering the trans-boundary
transport of the pollutants from non-Beijing emissions, the $O_3$ and $PM_{2.5}$ levels in Beijing are
remarkably increased, much higher than those in Mexico City, showing the important role of
trans-boundary transport in the air quality in Beijing. Hence, the cooperation with
neighboring provinces to decrease pollutant emissions is the optimum approach to mitigate
the air pollution in Beijing.
BTH has been considered as a polluted air basin (Zhao et al., 2009; Parrish et al., 2015).
However, although Beijing has implemented aggressive emission control strategies, it still
experiences $O_3$ and $PM_{2.5}$ pollutions during summertime, showing that the effective way to
improve air quality in Beijing is to reduce non-Beijing emissions in BTH. The FSA method is
based on simulations in which emissions from a certain region are completely turned on/off,
which can calculate the individual and synergistic contribution of local Beijing and

non-Beijing emissions by including or excluding the local or non-local emissions in this study. However, considering the nonlinear chemistry of $PM_{2.5}$ and $O_3$, especially regarding $O_3$ formation, the method might not well provide how the air quality is accurately when taking different emission reduction measures, and also emission reduction to zero in a vast region is apparently an infeasible scenario. This study mainly aims at providing a quantification of the effect of trans-boundary transport on the air quality in Beijing. Therefore, in the future study, sensitivity simulations of different emission reduction measures are needed to design reasonable emission control strategies.

It is worth noting that, although the WRF-CHEM model well captures the spatial distributions and temporal variations of pollutants, the model biases still exist. The discrepancies between the predictions and observations are possibly caused by the uncertainties in the emission inventory and the meteorological fields simulations (Zhang et al., 2015). Future studies need to be conducted to improve the WRF-CHEM model simulations, and further to assess the contributions of trans-boundary transport of emissions outside of Beijing to the air quality in Beijing, considering the rapid changes in anthropogenic emissions since implementation of the APPCAP. In addition, simulations for more pollution episodes should be investigated to evaluate the contribution of trans-boundary contributions to the air quality in Beijing for supporting the design and implementation of emission control strategies.



Data availability: The real-time $O_3$ and $PM_{2.5}$ are accessible for the public on the website
http://106.37.208.233:20035/. One can also access the historic profile of observed ambient
pollutants through visiting http://www.aqistudy.cn/.

*Acknowledgements.* This work was supported by the National Natural Science Foundation of
China (No. 41275153) and supported by the "Strategic Priority Research Program" of the
Chinese Academy of Sciences,Grant No. XDB05060500. Guohui Li is also supported by the
"Hundred Talents Program" of the Chinese Academy of Sciences. Naifang Bei is supported
by the National Natural Science Foundation of China (No. 41275101).

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

Table 1 Emissions of major anthropogenic species in July 2013 (Unit: $10^6$ g month$^{-1}$)

| Region | VOC | NO$_x$ | OC | SO$_2$ | CO | PM$_{2.5}$ |
|---|---|---|---|---|---|---|
| Beijing Municipality | 29303 | 26272 | 976 | 8796 | 119254 | 5319 |
| Tianjin Municipality | 29255 | 34534 | 1424 | 23204 | 181940 | 8831 |
| Hebei Province | 101710 | 190352 | 12732 | 136957 | 1239510 | 67877 |
| Shanxi Province | 35933 | 93069 | 6381 | 131758 | 355823 | 36473 |
| Shandong Province | 246538 | 235485 | 12181 | 246538 | 937528 | 77681 |


Table 2 Hourly mass concentrations of pollutants averaged in the afternoon at 12
monitoring sites in Beijing during summertime of 2013 and 2015.

| Pollutants | CO (mg m$^{-3}$) | SO$_2$ (μg m$^{-3}$) | NO$_2$ (μg m$^{-3}$) | O$_3$ (μg m$^{-3}$) | PM$_{2.5}$ (μg m$^{-3}$) |
|---|---|---|---|---|---|
| 2013 | 1.09 | 9.85 | 31.6 | 133.0 | 81.4 |
| 2015 | 0.88 | 5.71 | 23.6 | 163.2 | 61.9 |
| Change (%) | -20.0 | -42.0 | -25.1 | +22.8 | -24.0 |

Table 3 Average $O_3$ contributions (%) from 12:00 to 18:00 BJT in Beijing from local
emissions, non-Beijing emissions, the interactions of both emissions, and background from 5
to 14 July 2015.

| Emissions | Beijing | Surroundings | Interactions | Background |
|-----------|---------|--------------|--------------|------------|
| Date | $f'_B$ | $f'_S$ | $f'_{BS}$ | $f_0$ |
| 5 | 15.5 | 26.1 | -2.4 | 60.8 |
| 6 | 19.8 | 30.9 | -3.0 | 52.3 |
| 7 | 25.5 | 36.0 | -3.6 | 42.1 |
| 8 | 27.0 | 36.9 | -5.9 | 42.0 |
| 9 | 23.2 | 35.3 | -4.6 | 46.1 |
| 10 | 18.6 | 39.9 | -2.6 | 44.1 |
| 11 | 29.4 | 48.0 | -10.0 | 32.6 |
| 12 | 35.4 | 40.6 | -11.4 | 35.4 |
| 13 | 23.4 | 15.2 | -1.5 | 62.9 |
| 14 | 20.3 | 32.2 | -3.3 | 50.8 |
| Average | 22.4 | 36.6 | -5.1 | 46.1 |


Table 4 Average PM$_{2.5}$ contributions (%) in Beijing from local emissions, non-Beijing
emissions, the interactions of both emissions, and background from 5 to 14 July 2015.

| Emissions | Beijing | Surroundings | Interactions | Background |
|---|---|---|---|---|
| Date | $f'_B$ | $f'_S$ | $f'_{BS}$ | $f_0$ |
| 5 | 14.6 | 55.1 | 3.3 | 27.0 |
| 6 | 14.9 | 56.3 | 3.4 | 25.4 |
| 7 | 14.2 | 56.4 | 8.0 | 21.4 |
| 8 | 13.2 | 61.1 | 6.4 | 19.3 |
| 9 | 15.3 | 61.3 | 6.3 | 17.1 |
| 10 | 11.5 | 66.5 | 6.2 | 15.8 |
| 11 | 9.7 | 71.0 | 8.1 | 11.2 |
| 12 | 14.2 | 67.6 | 5.6 | 12.6 |
| 13 | 19.2 | 47.2 | 3.6 | 30.0 |
| 14 | 16.6 | 53.1 | 6.4 | 23.9 |
| Average | 13.7 | 61.5 | 5.9 | 18.9 |


Table 5 Aerosol species' contributions (%) from local emissions, non-Beijing emissions,
interactions of both emissions, and background, and mass fraction in the total PM$_{2.5}$ (%) in
Beijing averaged during the period from 5 to 14 July 2015.

| Emissions Species | Mass Fraction In Total PM$_{2.5}$ | Beijing $f'_B$ | Surroundings $f'_S$ | Interactions $f'_{BS}$ | Background $f_0$ |
|---|---|---|---|---|---|
| EC | 5.8 | 27.0 | 57.9 | 1.5 | 13.6 |
| POA | 9.8 | 20.8 | 49.0 | 5.3 | 24.9 |
| SOA | 25.0 | 14.6 | 64.2 | 5.9 | 15.3 |
| Ammonium | 13.7 | 14.5 | 65.7 | 1.5 | 18.3 |
| Nitrate | 14.1 | 10.1 | 71.7 | 18.1 | 0.1 |
| Sulfate | 25.1 | 6.5 | 52.9 | 3.4 | 37.2 |
| Unspecified | 6.5 | 21.2 | 61.4 | 1.6 | 15.8 |


**Figure Captions**

Figure 1 WRF-CHEM simulation domain. The blue circles represent centers of cities with ambient monitoring sites and the red circle denotes the NCNST site. The size of the blue circle denotes the number of ambient monitoring sites of cities.

Figure 2 Spatial distribution of anthropogenic (a) $NO_x$ (b) $VOC_s$ (c) OC (d) $SO_2$ emission rates (g month$^{-1}$) in the simulation domain.

Figure 3 Comparison of measured (black dots) and predicted (blue line) diurnal profiles of near-surface hourly (a) $PM_{2.5}$, (b) $O_3$, and (c) $NO_2$ averaged over all ambient monitoring stations in Beijing from 5 to 14 July 2015.

Figure 4 Comparison of measured (black dots) and simulated (black line) diurnal profiles of submicron aerosol species of (a) POA, (b) SOA, (c) sulfate, (d) nitrate, and (e) ammonium at NCNST site in Beijing from 5 to 14 July 2015.

Figure 5 Comparison of measured (black dots) and predicted (blue line) diurnal profiles of near-surface hourly (a) $PM_{2.5}$, (b) $O_3$, and (c) $NO_2$ averaged over all ambient monitoring stations in BTH from 5 to 14 July 2015.

Figure 6 Pattern comparison of simulated vs. observed near-surface $PM_{2.5}$ at 10:00 BJT during the selected periods from 5 to 14 July 2015. Colored circles: $PM_{2.5}$ observations; color contour: $PM_{2.5}$ simulations; black arrows: simulated surface winds.

Figure 7 Same as Figure 6, but for $O_3$ at 15:00 BJT.

Figure 8 Same as Figure 6, but for $NO_2$ at 08:00 BJT.

Figure 9 Temporal variations of total net horizontal transport flux of $PM_{2.5}$, $O_3$ and $NO_2$ over Beijing boundary (blue line) and the contribution of non-Beijing emission to the $PM_{2.5}$, $O_3$ and $NO_2$ concentrations in Beijing (black line) during the study episode.

Figure 10 Temporal variations of the average near-surface $O_3$ and $PM_{2.5}$ concentrations from $f_{BS}$ with all the emissions (black line), $f_B$ with Beijing emissions alone (blue line), and $f_S$ with non-Beijing emissions alone (red line) in Beijing from 5 to 14 July 2015.

Figure 11 Temporal variations of the average contributions to the near-surface aerosol species concentrations from total emissions (black line, defined as $f_{BS}$), local emissions (blue line, $f'_B$, defined as $f_B - f_0$), non-Beijing emissions (red line, $f'_S$, defined as $f_S - f_0$), the emission interactions (green line, $f'_{BS}$, defined as $f_{BS} - f_B - f_S + f_0$), and background (black dashed line, defined as $f_0$) in

**972**     Beijing from 5 to 14 July 2015.

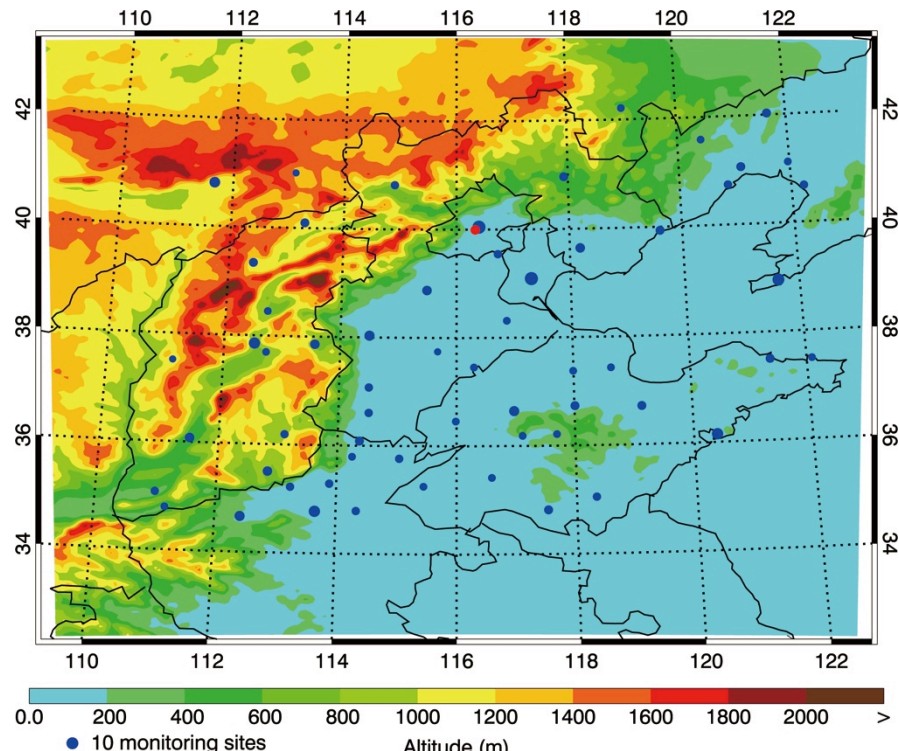

**976** Figure 1

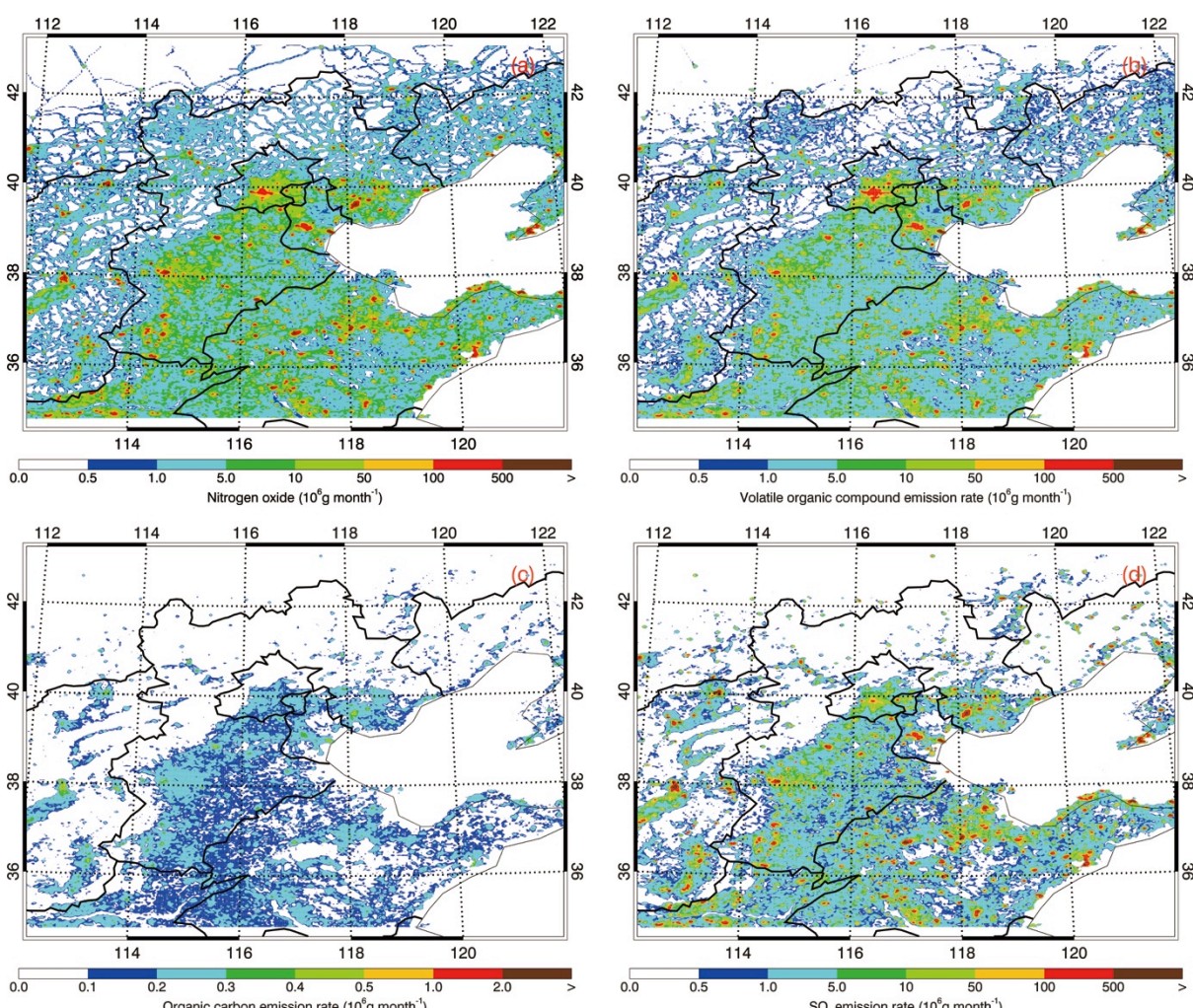



Figure 2


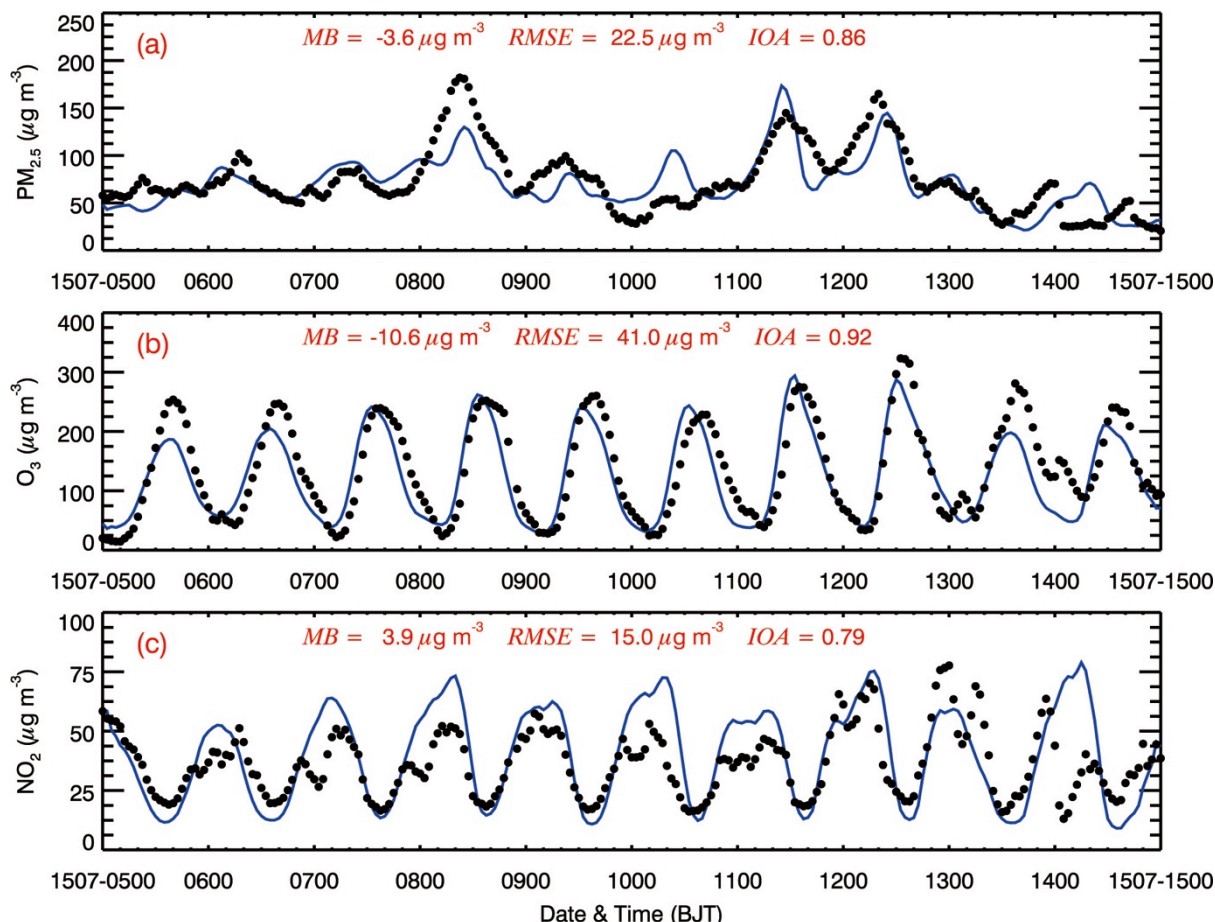



Figure 3




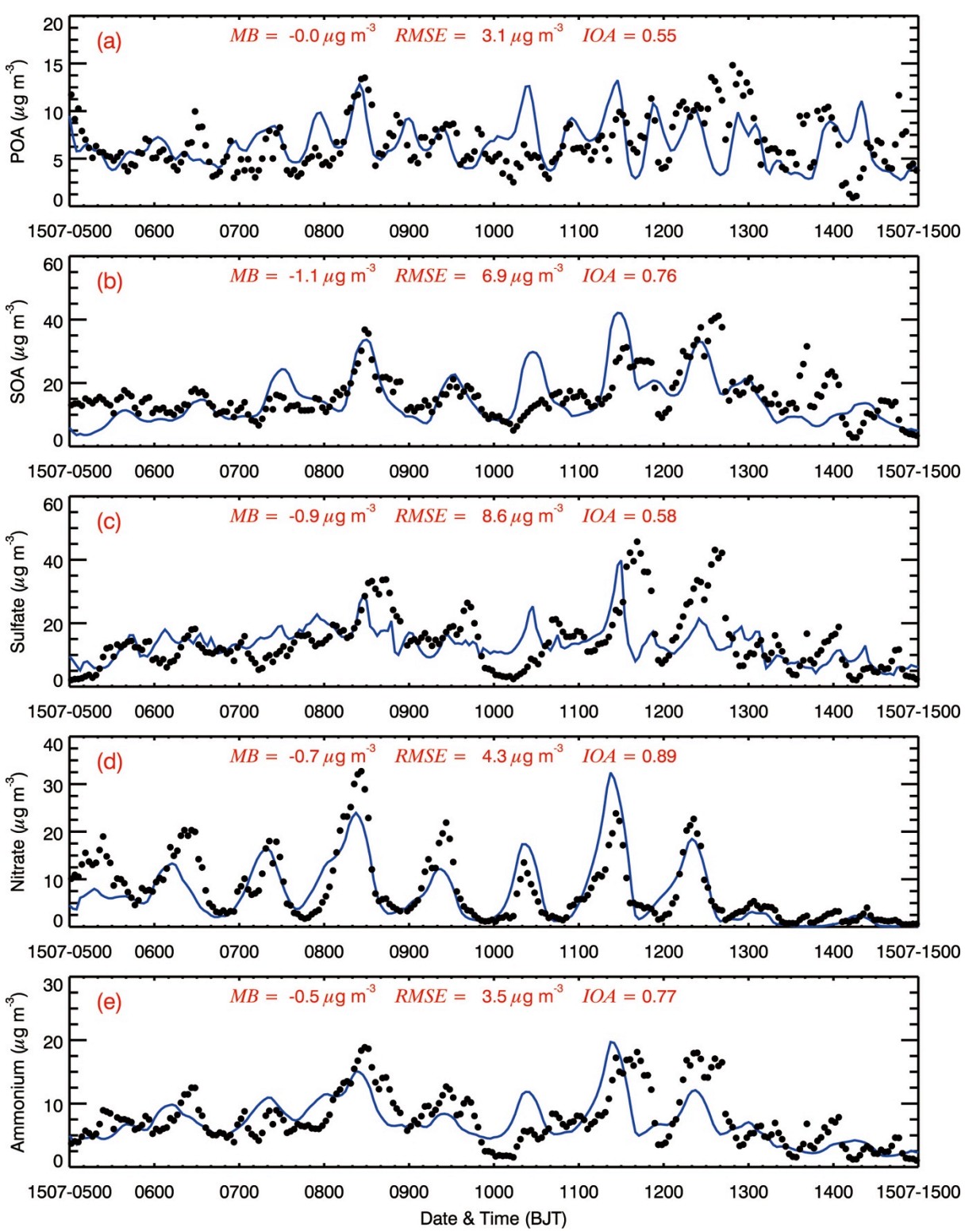

**990**

**991**

**992**   Figure 4

**993**

**994**

**995**

**996**

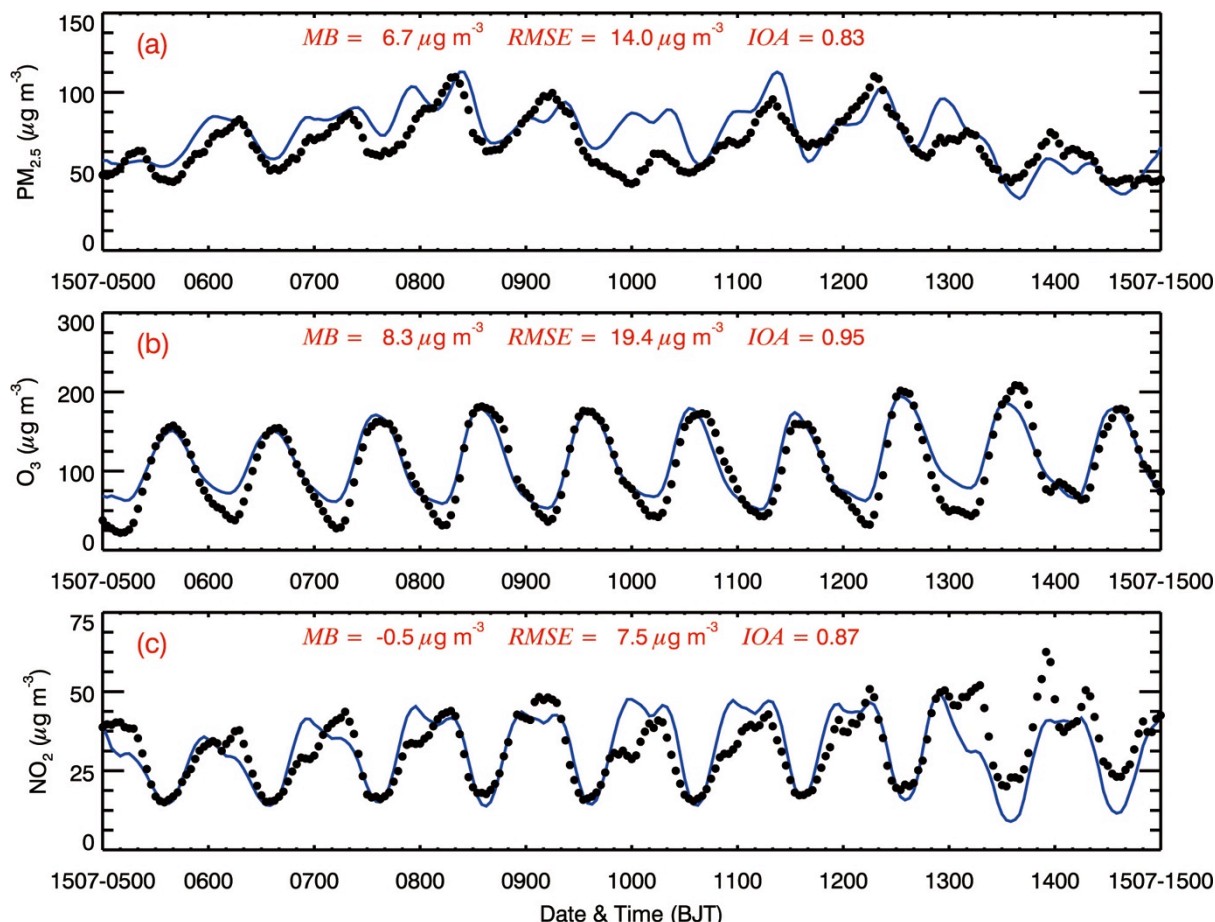



Figure 5




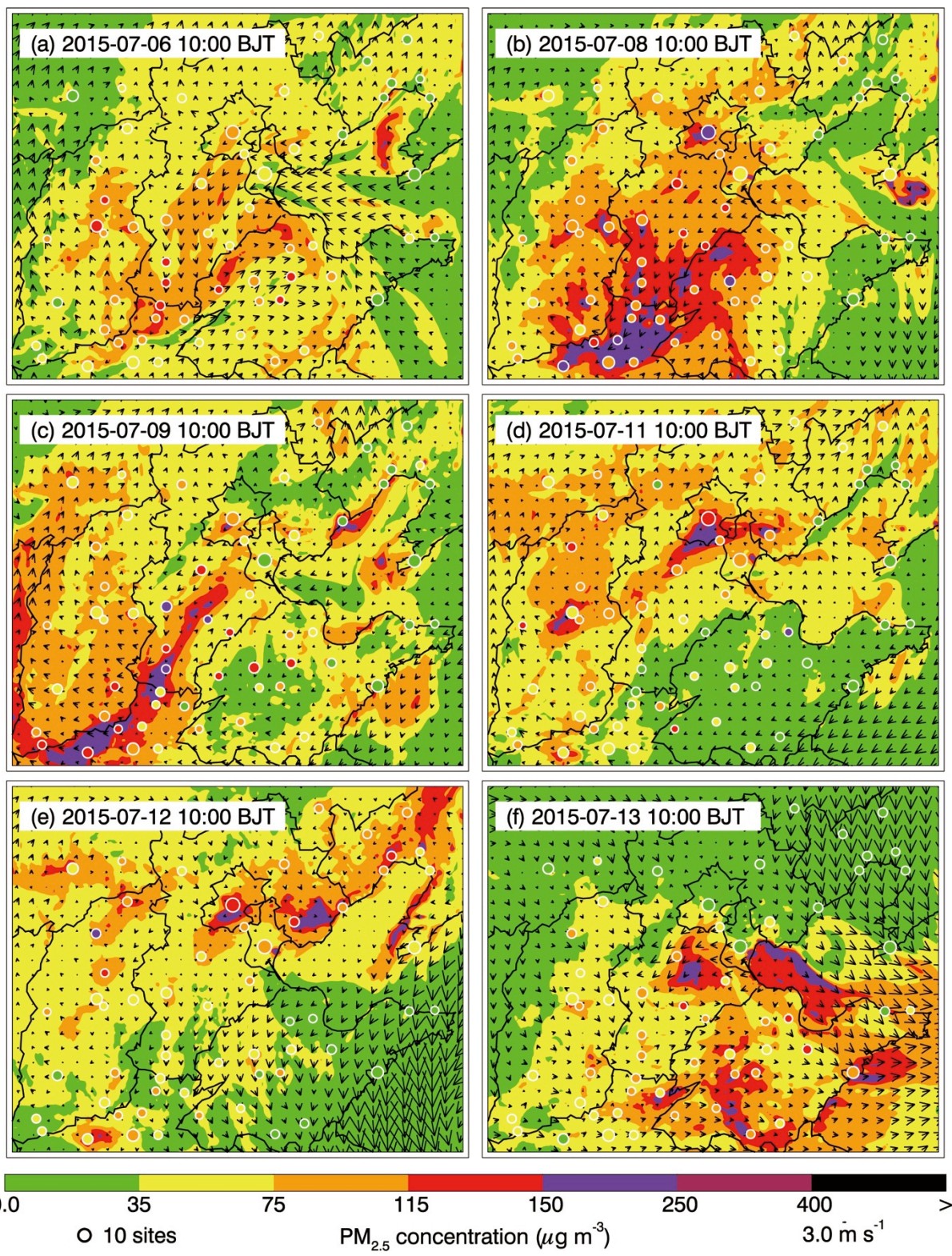

**1004**

**1005**

**1006**    Figure 6

**1007**

**1008**

**1009**

**1010**

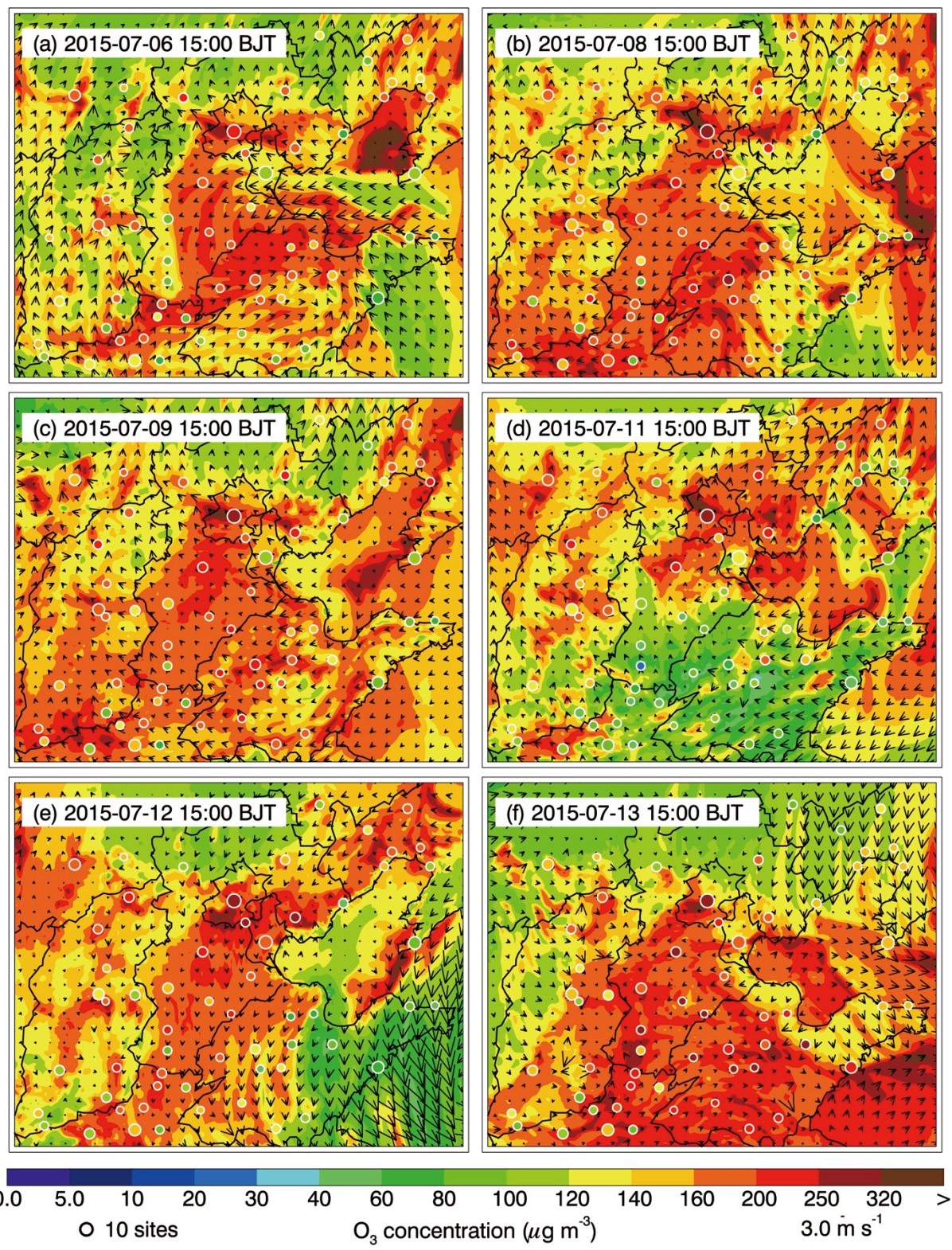



Figure 7




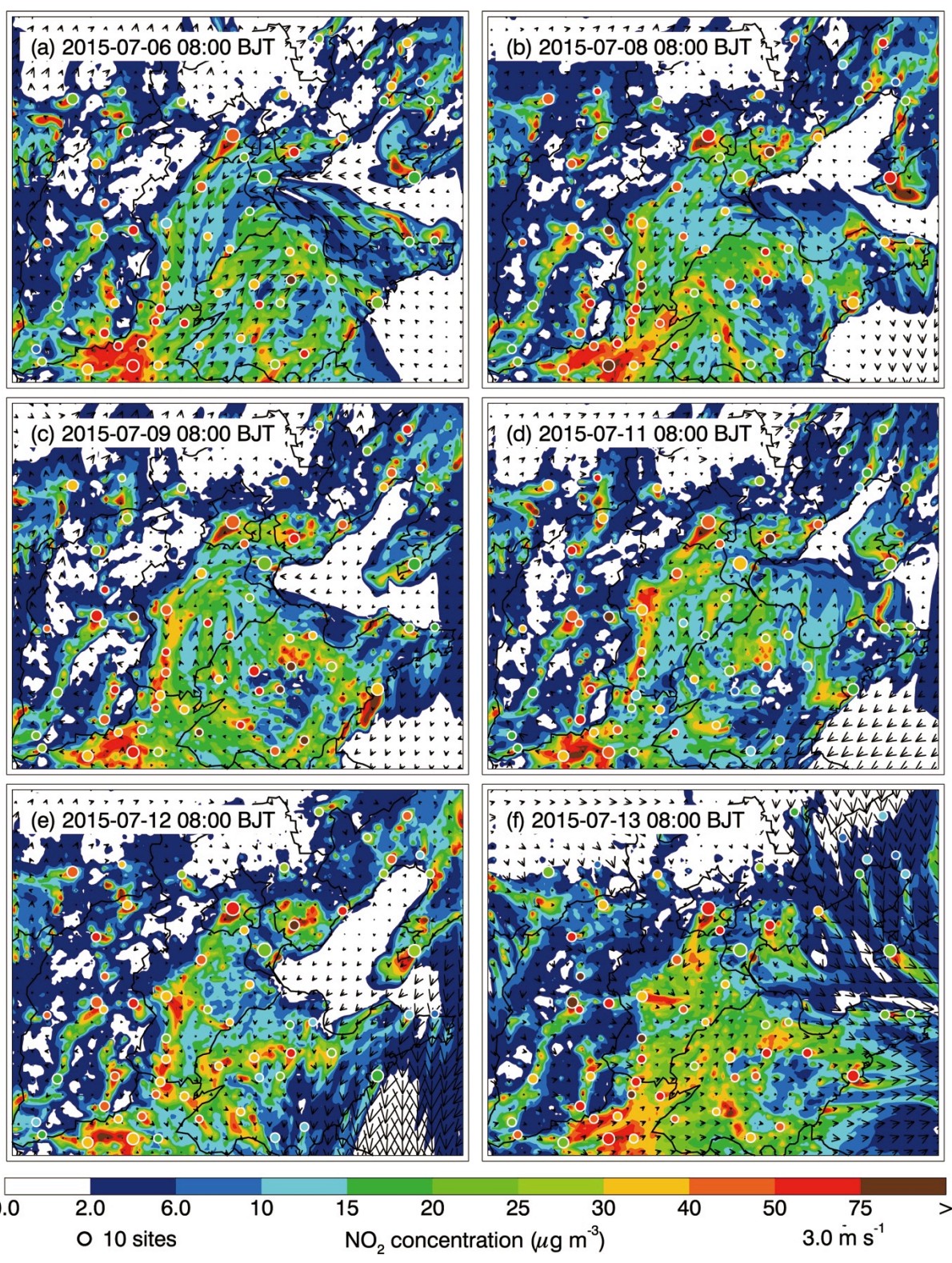

Figure 8

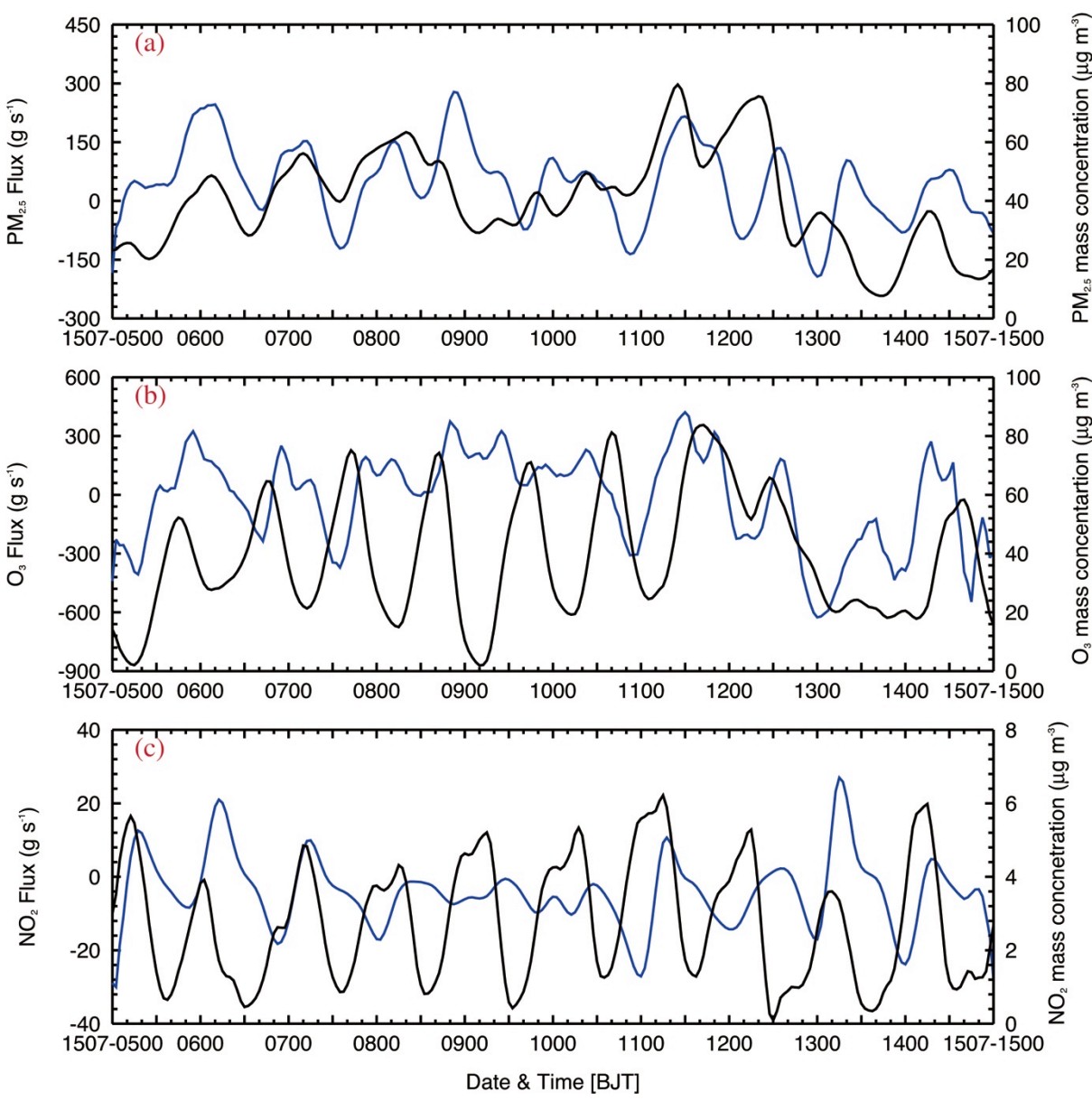

Figure 9

**1026**

**1027**

**1028**

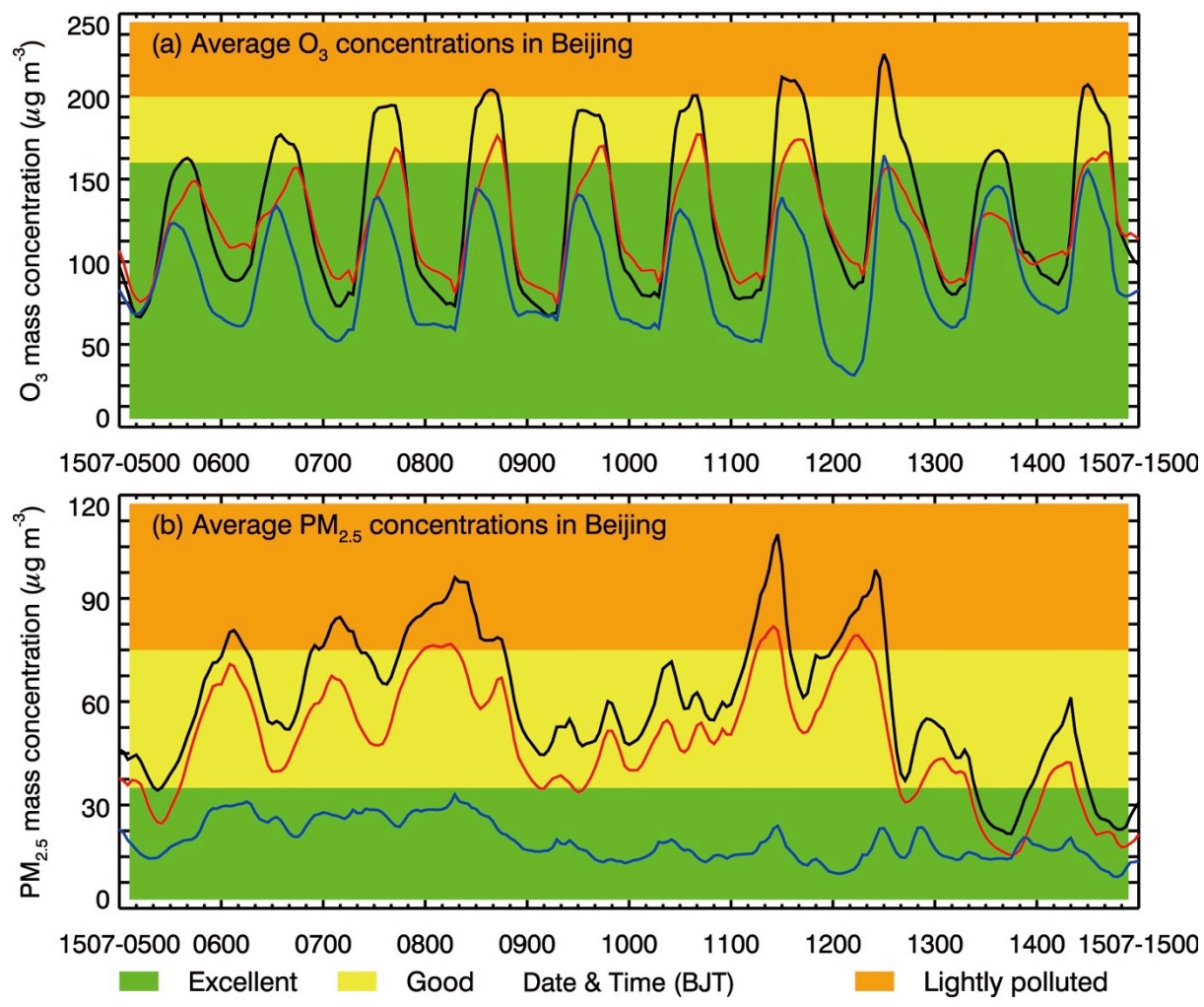

**1029**

**1030**

**1031**    Figure 10

**1032**

**1033**

**1034**

**1035**

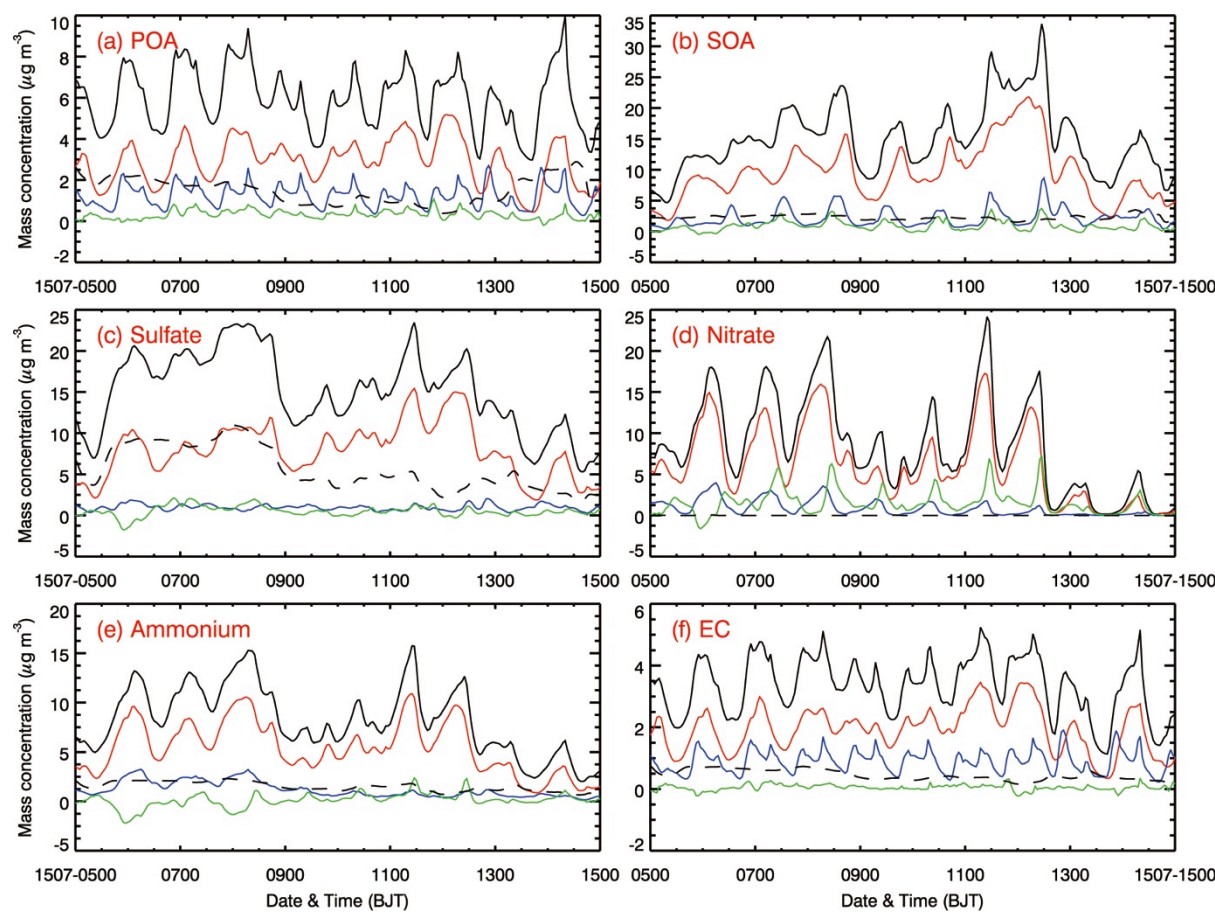



Figure 11


