# Peer review of "Contributions of Trans-boundary Transport to Summertime Air Quality in Beijing, China"

_Atmospheric Chemistry and Physics, 2016_

## Referee Comment (RC1) · Anonymous Referee #1 · 13 Aug 2016

The authors employed the WRF-CHEM model to assess the contributions of trans-boundary transport to the air quality in Beijing during a persistent air pollution episode from 5 to 14 July 2015 in Beijing-Tianjin-Hebei (BTH), China. They showed that the WRF-CHEM model reproduced well the temporal variations of aerosol species compared to measurements in Beijing. The authors indicated a large contribution from the regional transport and suggested that the cooperation with neighboring provinces to mitigate pollutant emissions is a key for Beijing to improve air quality. This is a good work to investigate the air pollution problem in China, and the paper was reasonably well written. As such, it can be published in ACPD.
* * *

---

## Referee Comment (RC2) · Anonymous Referee #2 · 5 Sep 2016

The authors used the WRF-CHEM model to investigate the contributions of the non-local emissions to the summertime air pollution episode in Beijing. By turning on/off the local and/or non-local emissions and using the factor separation method, the authors found that during the episode (July 5 to 14, 2015), non-Beijing emissions contributed dominantly over local emissions to the $PM_{2.5}$ and ozone concentrations in Beijing. Although previous studies have discussed the topic of local vs. non-local contributions in Beijing, this paper provided updated information on the topic by focusing on the summer of 2015, after Beijing has taken drastic emission control measures in recent years. My major criticism, however, is that the paper lacks sufficient quantitative analysis and scientific discussion as a research paper. Therefore, I recommend a major revision before publication.

**Major comments**

Note that I make a bunch of suggestions in the major comments. I do not think it is necessary for the authors to follow these suggestions completely but I do think the major questions or concerns should be addressed in the revision.

1. The authors gave a lengthy description of model evolution of $PM_{2.5}$ and ozone in the BTH region during the episode (Line 257-301, Fig. 5-9). However, these descriptions are mostly qualitative and are not scientifically insightful. The multi-panel figures (Fig. 5-9) are too complex to help a reader understand what the authors say in the text. I would suggest the author to rewrite the section and keep it succinct. It may also be a good idea to move some of the content to a supplement.

   In addition, Fig. 5-8 shows that $PM_{2.5}$ and $O_3$ in BTH are being transported by wind. However, this is not equivalent to the contribution of non-local emissions. First, $PM_{2.5}$ and $O_3$

in Beijing can also be contributed by local productions from precursors emitted outside Beijing. Second, it is also possible that Beijing emissions can contribute to the production of $PM_{2.5}$ and $O_3$ outside Beijing, and then these $PM_{2.5}$ and $O_3$ are transported back to the city. Since the authors did not provide quantitative analysis to rule out these possibilities, Fig. 5-8 cannot support the author's conclusion very well.  Instead of describing the 12-panel figures, I would suggest the author to do more quantitative analysis (for example, diagnosis of the flux of $PM_{2.5}$ and $O_3$ across the city boundary) and discuss whether precursor transport is important.

2.  For the purpose of this paper, the accuracy of the emission inventory is very important. However, the description of the emission inventory in the manuscript is too brief. What year does the emission inventory based upon? Is it for the year 2006 as in Zhang et al. (2009) or is it updated? What are emissions from Beijing compared with those from the surrounding regions? How uncertain is the emission inventory? This information are essential for a reader to assess the significance of the paper's conclusion.

   In addition, as the authors pointed out, the emissions have been greatly reduced in Beijing in recent years. Therefore, if emission inventory for multiple years are available, it would be very interesting to conduct additional simulations and calculations and see how the implementation of APPCAP affects the contributions of trans-boundary transport.

3.  The authors used the FSA method to separate the impact of local and non-local emissions.  I have several questions about the results that the authors present.

a. In Table 2, the background contribution $f_0$ varies from 32.6 ppbv to 62.9 ppbv. Why does background vary so much? In addition, $f_0$ and $f_s$' (surrounding) anti-correlates very well ($R^2$=0.89 based on my calculation). In Table 3, $f_0$ and $f_s$' also anti-correlates really well ($R^2$=0.92). Why is that? The author should give more discussion to provide insight into these interesting results.

b. In Table 2 and Figure 10(a). The authors show that local contribution to $O_3$ is much less than non-local contributions. Beijing, with so much traffic, should have large amount of $NO_x$ emissions. Given $NO_x$ lifetime in the summer should be on the order of several hours, regional transport of $NO_x$ should not be very significant. So the question follows, why is regional contribution so larger? Is the input of regional $O_3$ or input of precursors? If it's the input of $O_3$, why do regions surrounding Beijing have such high $O_3$ production? Do they emit a lot of $NO_x$ and VOCs. The authors should discuss the matter by referring to emission inventory (See major comment 2) and flux diagnosis (See major comment 1).

c. In Table 3 and Figure 10(b), the authors show somewhat counter-intuitive results that non-local emissions almost always contribute more than 50% of Beijing and $f_s$' follows the $PM_{2.5}$ concentration perfectly. In my opinion, the authors should show additional results using emission inventory, flux diagnosis, etc. to convince a reader that their calculation is right and consistent. Or, it may be a good idea for the authors to show simulation and FSA results outside an episode in the same summer to give readers a sense of how the "control" case looks like.

d. More fundamentally, I am concerned that the FSA results in the paper are somewhat misleading to readers because in the four simulations to derive FSA, the author turn on/off the local or non-local emissions completely. As a result of the nonlinear chemistry, these

simulations cannot give accurate information about the local sensitivity of air quality to emission reduction. Theoretically, it is possible that reducing local emissions may still be more effective than reducing non-local emissions. I suggest the authors to make clear in the text about the limitation of their method.

4. The paper also lacks sufficient discussion of the results in the context of previous studies. The authors mentioned several previous studies on local vs. non-local emissions. The result of this paper stands out as reporting most significant non-local contributions. The paper will be much better if the authors can discuss their paper in context of these studies (in terms of method, results, discrepancies, or agreement, etc.).

**Minor comments**

5. Abstract. The words "pure local emissions" and "pure emissions outside Beijing" are confusing. Are there impure emissions? I suggest to use "local emissions", "non-local (or non-Beijing) emissions", and "interactions between local and non-local emissions".

6. Line 121. Is NCEP reanalysis only used for boundary and initial conditions? Is WRF configured to nudge the meteorology fields towards reanalysis?

7. Section 3.1. This section is only a summary of changes in air quality of Beijing during recent years, which are background information rather than research results. Therefore this section should not be within "results and discussion".

8. Line 217-219, 243-238, 290. The authors attributes the model errors to inability of WRF-CHEM to resolve convection in several occasions. However, these statements are not backed by any data (e.g., observations of convective clouds etc.). Given there are so many plausible error sources in a 3-D chemical transport model, I would suggest not to make guesses on

what leads to the model errors (emission errors, meteorology errors, etc.), unless supported by observation data or model sensitivity test.

9. Line 250. The title number should be 3.2.3

---

## Referee Comment (RC3) · Anonymous Referee #1 · 6 Sep 2016

The authors employed the WRF-CHEM model to assess the contributions of trans-boundary transport to air quality in Beijing during a persistent air pollution episode from 5 to 14 July 2015 in Beijing-Tianjin-Hebei (BTH), China. They showed that the WRF-CHEM model reproduced well the temporal variations of the aerosol species compared to the measurements in Beijing. The authors indicated a large contribution from the regional transport and suggested that a coordinated mitigation for pollutant emissions with neighboring provinces is key to improve air quality in Beijing. This is a good work to investigate the air pollution problem in China and, in particular, provides quantitative insight into the contributions of trans-boundary transport of outside emissions the PM2.5 and O3 levels in Beijing. The paper was in general reasonably well written, although it could be benefitted from additional editing. I recommend publication of this work, after a revision to address the following issues.

Their last statement in the abstract sounded to be out-placed and non-substantiated. It has been commonly well established that the accuracy of simulations by chemical transport models (such as WRF-CHEM) is largely dependent of several factors, including emission inventory, chemistry, and meteorological fields (including the PBL height) (e.g., Zhang et al., Formation of urban fine particulate matter, Chem. Rev. 115, 3803, 2015). What was really missing from this manuscript is a careful account of those various factors that impact their simulations and conclusions.

For example, the different VOC types exhibit distinct kinetic behaviors, and their contributions to O3 and SOA formation are also distinct (e.g., Suh et al., Oxidation mechanism of aromatic peroxy and bicyclic radicals from OH-toluene reactions, J. Am. Chem. Soc. 125, 12655, 2003; Fan et al., Atmospheric oxidation mechanism of isoprene, Environ. Chem. 1, 140, 2004). How well was the VOC EI represented in the model, and how did the VOC EI uncertainty impact their simulations?

The PM problem in Beijing has been well characterized by efficient and rapid secondary formation (Guo et al., Elucidating severe urban haze formation in China, Proc. Natl. Acad. Sci. USA 111, 17373, 2014; Zhang et al., Insufficient evidence for the contribution of regional transport to severe haze formation in Beijing, Proc. Natl. Acad. Sci. USA 112, E2741, doi:10.1073/pnas.1503855112, 2015). How well did their model handle those secondary PM formation processes, including nucleation and growth from the various organic and inorganic species (Fan et al., Contribution of secondary condensable organics to new particle formation: A case study in Houston, Texas, Geophys. Res. Lett. 33, L15802, doi:10.1029/2006GL026295, 2006). How well did their model handle the particle-phase reactions, including those associated with small di-carbonyls (glyoxal and methyl glyoxal) that could be particularly important for urban PM formation because of their traffic origin (Zhao et al., Heterogeneous reactions of methylglyoxal in acidic media: Implications for secondary organic aerosol formation, Environ. Sci. Technol. 40, 7682, 2006; Gomez et al., Heterogeneous chemistry of glyoxal on acidic solutions – An oligomerization pathway for secondary organic aerosol formation, J.

Phys. Chem. 118, 4457, 2015).

Another problematic area was related to the MET part, including PBL. In particular, it has become clear that the PBL-pollution interaction plays a key role in pollutant accumulation in Beijing (Wang et al., Light absorbing aerosols and their atmospheric impacts, Atmos. Environ. 81, 713, 2013; Peng et al., Markedly enhanced absorption and direct radiative forcing of black carbon under polluted urban environments, Proc. Natl. Acad. Sci. USA 113, 4266, 2016).

The manuscript contained some stylistic/grammatical/typographic errors, which hindered its readability. Here are a few examples. In the title, delete "the". Line 19, replace "emissions outside of Beijing" by "outside emissions" Line 23 and other places, the word "pure" is ambiguous and can be deleted or replaced by "the local-only emissions". Line21-22, "pure emissions outside of Beijing" to "the outside emissions". Line 23, "local emissions with those outside of Beijing" to "local and outside emissions". Line 23, "The pure emissions outside of Beijing" to "The outside emissions". Line 25, "pure Beijing local emissions" to "the local emissions". Line 25, "The emissions interactions" to "The interactions between local and outside emissions". Line 31, "need to be performed to improve" to "are needed to improve".

---

## Referee Comment (RC4) · Anonymous Referee #3 · 29 Sep 2016

Review on "Contributions of Trans-boundary Transport to the Summertime Air Quality in Beijing, China" by Wu et al.

General comments

This manuscript presents a WRF-CHEM modeling study using to evaluate the contribution of regional emissions to the air quality in Beijing during a summertime pollution episode using the FSA analysis. This study comes timely as there have been debates over whether local emissions play the major contribution to the air pollution in Beijing. The methodology is sound, and the results are well presented and organized. I would recommend it for publishing with a few minor revisions.

Specific comments

1. P1 lines 27-28 and P21 lines 462-463, it is a big jump to extrapolate the results from an episode to the whole summer season. You need to prove that the episode studied is representative of the summertime situation in Beijing. In addition, air quality "primarily determined by the trans-boundary transport" may only be applied to PM2.5, not O3, since background O3 accounts for 46% of the afternoon O3 (line 338). Maybe I have a misunderstanding in here, which is related to the comment below.

2. Confusions on some concepts. To my understanding, the "trans-boundary transport" term refers to the transport of regional anthropogenic emissions (i.e., $f_S$), and it does not include "background" ($f_0$). Am I right? In addition, in the FSA analysis, which term include the biogenic emissions? Are the interactions between anthropogenic and biogenic emissions accounted? Does the trans-boundary transport include the impacts of biogenic emissions?

3. In both the abstract and summary sections, besides considering the uncertainties in emissions and meteorology, simulations for more pollution episodes should be addressed to "evaluate trans-boundary transport contributions to the air quality in Beijing for supporting the design and implementation of emission control strategies".

4. P2-3,lines 55-56, "daily average of up to 110…", daily average or daytime average?

5. P5 lines 111-113, more descriptions of the episode are needed, including meteorological conditions, which may help to add information whether or not the episode is representative of the summertime air pollution in Beijing. Also, what does the "mean daily" mean here? episode average?

6. P9 line 191-193, the differences in CO, SO2, NOx and PM2.5 between 2013 and 2015 are attributed solely to the emission change. Are the meteorological conditions similar between these two periods?

7. P10-11 Section 3.2, $NO_3^-$ and $NH_4^+$ are shown in Figure 3, but there are no discussions or descriptions of these two components. Point comparisons may also contribute to the biases.

8. Figures 10 and 11, regarding the contributions from total emissions, emissions from Beijing, and emissions outside Beijing. As I understand, the last two terms are calculated as $f_B' + f_{BS}'$ ($or\ f_{BS} - f_S$) and $f_S' + f_{BS}'$ ($or\ f_{BS} - f_B$), but how is the first term (contribution from total emissions) calculated? $f_{BS} - f_0$? or sum of the last two terms? Captions in Fig. 10 and 11 are a bit of confusing: $f_B$, $f_S$, and $f_{BS}$ represent simulation results, not contributions.

9. P15 line 337, 45.6%, but in table 2, the number is 46.1%.

10.     Table 3 is shown and is not discussed.

Technical corrections

1. Line 25, change "more" to "higher"
2. Line 32, change "reasonably" to "better"
3. Line 134, Stein et al. (1993) should be Stein and Alpert (1993)
4. Line 189, delete "hourly"
5. Line 227, change "the failure of" to "biased"
6. Lines 533-538, the two references do not show in the text.

---

## Author Comment (AC1) · 17 Nov 2016

**Reply to Anonymous Referee #1**

We thank the reviewer for the careful reading of the manuscript and helpful comments. We have revised the manuscript following the suggestion, as described below.

The authors employed the WRF-CHEM model to assess the contributions of trans- boundary transport to air quality in Beijing during a persistent air pollution episode from 5 to 14 July 2015 in Beijing-Tianjin-Hebei (BTH), China. They showed that the WRF- CHEM model reproduced well the temporal variations of the aerosol species compared to the measurements in Beijing. The authors indicated a large contribution from the regional transport and suggested that a coordinated mitigation for pollutant emissions with neighboring provinces is key to improve air quality in Beijing. This is a good work to investigate the air pollution problem in China and, in particular, provides quantitative insight into the contributions of trans-boundary transport of outside emissions the $PM_{2.5}$ and $O_3$ levels in Beijing. The paper was in general reasonably well written, although it could be benefitted from additional editing. I recommend publication of this work, after a revision to address the following issues.

**1 Comment:** Their last statement in the abstract sounded to be out-placed and non-substantiated. It has been commonly well established that the accuracy of simulations by chemical transport models (such as WRF-CHEM) is largely dependent of several factors, including emission inventory, chemistry, and meteorological fields (including the PBL height) (e.g., Zhang et al., Formation of urban fine particulate matter, Chem. Rev. 115, 3803, 2015). What was really missing from this manuscript is a careful account of those various factors that impact their simulations and conclusions.

**Response**: We have removed the sentence from the abstract and included following discussions in the conclusion: *"The discrepancies between the predictions and observations are possibly caused by the uncertainties in the emission inventory, chemistry, and the meteorological fields simulations (Zhang et al., 2015)."*.

**2 Comment:** For example, the different VOC types exhibit distinct kinetic behaviors, and their contributions to $O_3$ and SOA formation are also distinct (e.g., Suh et al., Oxidation mechanism of aromatic peroxy and bicyclic radicals from OH-toluene reactions, J. Am. Chem. Soc. 125, 12655,

2003; Fan et al., Atmospheric oxidation mechanism of isoprene, Environ. Chem. 1, 140, 2004).
How well was the VOC EI represented in the model, and how did the VOC EI uncertainty
impact their simulations?

**Response:** We have not provided quantitative evaluations of VOCs simulations in the
manuscript due to lack of VOC measurements. We have included the following discussions in
Section 2.2: *"For example, different VOCs types exhibit distinct kinetic behaviors, and as an
important fraction of total VOCs in the urban atmosphere, aromatics are responsible for the
photochemical $O_3$ production and secondary organic aerosol formation (Suh et al., 2003; Fan et
al., 2004). In the SAPRC99, aromatics are lumped into ARO1 and ARO2. ARO1 mainly includes
toluene, benzene, ethylbenzene, and other aromatics with the reaction rate with OH (kOH) less
than $2 \times 10^4$ $ppm^{-1}$ $min^{-1}$. ARO2 includes xylene, trimethylbenzene, and other aromatics with kOH
greater than $2 \times 10^4$ $ppm^{-1}$ $min^{-1}$. Additionally, biogenic VOCs also play a considerable role in
the $O_3$ production (Li et al., 2007), and monoterpenes and isoprene are the main biogenic VOCs
in the SAPRC99 chemical mechanism."*

**3 Comment:** The PM problem in Beijing has been well characterized by efficient and rapid
secondary formation (Guo et al., Elucidating severe urban haze formation in China, Proc. Natl.
Acad. Sci. USA 111, 17373, 2014; Zhang et al., Insufficient evidence for the contribution of
regional transport to severe haze formation in Beijing, Proc. Natl. Acad. Sci. USA 112, E2741,
doi:10.1073/pnas.1503855112, 2015). How well did their model handle those secondary PM
formation processes, including nucleation and growth from the various organic and inorganic
species (Fan et al., Contribution of secondary condensable organics to new particle formation: A
case study in Houston, Texas, Geophys. Res. Lett. 33, L15802, doi:10.1029/2006GL026295,
2006)

**Response:** We have included a paragraph about the secondary PM formation process in Section
2.1:*"The aerosol component of the Community Multiscale Air Quality (CMAQ) model is
designed to be an efficient and economical depiction of aerosol dynamics in the atmosphere
(Binkowski and Roselle, 2003). The particle size distribution in the study is represented as the
superposition of three lognormal subdistributions, called modes, which includes the processes of
coagulation, particle growth by the addition of mass, and new particle formation. Following the*

*work of Kulmala et al. (1998), the new particle production rate presented here is calculated as a parameterized function of temperature, relative humidity, and the vapor-phase $H_2SO_4$ concentration due to binary nucleation of $H_2SO_4$ and $H_2O$ vapor, and the new particles are assumed to be 2.0 nm diameter. A number of recent studies have shown that organic compounds can play an important role in the nucleation process (Zhang et al., 2009, 2012, 2015). The contribution from organic acids likely explains the high level of aerosols, especially in polluted urban area, where large amount of organic acids can be emitted directly and produced by photochemical oxidation of hydrocarbons (Fan et al., 2006), which needs to be considered in the further study."*

**4 Comment:** How well did their model handle the particle-phase reactions, including those associated with small di-carbonyls (glyoxal and methyl glyoxal) that could be particularly important for urban PM formation because of their traffic origin (Zhao et al., Heterogeneous reactions of methylglyoxal in acidic media: Implications for secondary organic aerosol formation, Environ. Sci. Technol. 40, 7682, 2006; Gomez et al., Heterogeneous chemistry of glyoxal on acidic solutions – An oligomerization pathway for secondary organic aerosol formation, J. Phys. Chem. 118, 4457, 2015).

**Response:** In the SOA module, we have included the contribution of glyxal and methylglyoxal to the SOA formation, and we have clarified in Section 2.1: *"Recent studies have shown that small di-carbonyls (glyoxal and methylglyoxal) are important for the aerosol formation due to their traffic origin (Zhao et al., 2006; Gomez et al., 2015). Li et al. (2011a) have indicated that glyoxal and methylglyoxal can contribute about 10% of the SOA in the urban area of Mexico City. The SOA formation from glyoxal and methylglyoxal in this study is parameterized as a first-order irreversible uptake by aerosol particles and cloud droplets, with a reactive uptake coefficient of $3.7 \times 10^{-3}$ for glyoxal and methylglyoxal (Zhao et al., 2006; Volkamer et al., 2007; Gomez et al., 2015)."*

**5 Comment:** Another problematic area was related to the MET part, including PBL. In particular, it has become clear that the PBL-pollution interaction plays a key role in pollutant accumulation in Beijing (Wang et al., Light absorbing aerosols and their atmospheric impacts, Atmos. Environ.

81, 713, 2013; Peng et al., Markedly enhanced absorption and direct radiative forcing of black carbon under polluted urban environments, Proc. Natl. Acad. Sci. USA 113, 4266, 2016).

**Response:** We have considered the role of PBL-interaction in pollutant accumulation in the manuscript. We have clarified in Section 3.2.2: *"It is clear that the PBL-pollution interaction plays an important role in the pollutant accumulation in Beijing (Wang et al., 2013;Peng et al., 2016). Mixing of Beijing local emissions with those outside of Beijing increases the aerosol concentrations in the PBL and decreases the incoming solar radiation down to the surface, cooling the temperature of the low level atmosphere to suppress the development of PBL and hinder the aerosol dispersion in the vertical direction."*

**Stylistic/grammatical/typographic errors**

(1) We have removed "the" in the title.

(2) We have replaced "emissions outside of Beijing" by "outside emissions" or "non-Beijing" emissions in the manuscript.

(3) The word "pure" in the manuscript has been removed.

(4) We have changed "local emissions with those outside of Beijing" to "local and outside emissions".

(5) We have changed "The pure emissions outside of Beijing" to "The outside emissions", and "pure Beijing local emissions" to "the local emissions".

(7) We have changed "The emissions interactions" to "The interactions between local and outside emissions".

(8) We have changed "need to be performed to improve" to "are needed to improve".

---

## Author Comment (AC2) · 17 Nov 2016

**Reply to Anonymous Referee #2**

We thank the reviewer for the careful reading of the manuscript and helpful comments. We have revised the manuscript following the suggestion, as described below.

The authors used the WRF-CHEM model to investigate the contributions of the non-local emissions to the summertime air pollution episode in Beijing. By turning on/off the local and/or non-local emissions and using the factor separation method, the authors found that during the episode (July 5 to 14, 2015), non-Beijing emissions contributed dominantly over local emissions to the $PM_{2.5}$ and ozone concentrations in Beijing. Although previous studies have discussed the topic of local vs. non-local contributions in Beijing, this paper provided updated information on the topic by focusing on the summer of 2015, after Beijing has taken drastic emission control measures in recent years. My major criticism, however, is that the paper lacks sufficient quantitative analysis and scientific discussion as a research paper. Therefore, I recommend a major revision before publication.

**Major comments**

Note that I make a bunch of suggestions in the major comments. I do not think it is necessary for the authors to follow these suggestions completely but I do think the major questions or concerns should be addressed in the revision.

**1 Comment:** The author gave a lengthy description of model evolution of $PM_{2.5}$ and ozone in the BTH region during the episode (Line 257-301, Fig. 5-9). However, these descriptions are mostly qualitative and are not scientifically insightful. The multi-panel figures (Fig. 5-9) are too complex to help a reader understand what the authors say in the text. I would suggest the author to rewrite the section and keep it succinct. It may also be a good idea to move some of the content to a supplement.

**Response:** We have moved Figure 6, Figure 7, and the corresponding analysis to Supplementary Information Section SI-1.

**2 Comment:** In addition, Fig. 5-8 shows that $PM_{2.5}$ and $O_3$ in BTH are being transported by wind. However, this is not equivalent to the contribution of non-local emissions. First, $PM_{2.5}$ and $O_3$ in Beijing can also be contributed by local productions from precursors emitted outside Beijing. Second, it is also possible that Beijing emissions can contribute to the production of $PM_{2.5}$ and $O_3$ outside Beijing, and then these $PM_{2.5}$ and $O_3$ are transported back to the city. Since the authors did not provide quantitative analysis to rule out these possibilities, Fig. 5-8 cannot support the author's conclusion very well. Instead of describing the 12-panel figures, I would suggest the author to do more quantitative analysis (for example, diagnosis of the flux of $PM_{2.5}$ and $O_3$ across the city boundary) and discuss whether precursor transport is important.

**Response:** We have added a figure (Figure 9) and a table (SI-Table 1) to provide the quantitative analysis of the flux of $PM_{2.5}$, $O_3$, and $NO_2$ across the city boundary to highlight the importance of trans-boundary transport of pollutants. We have added a section 3.2.1 to clarify the horizontal transport:

*"**3.2.1 Analysis of Horizontal Transport of $O_3$ and $PM_{2.5}$**

The analysis in Section 3.1.3 has shown the strong correlation between the airflow and the high level of pollutants in Beijing during the study episode. It is essential to confirm whether the continuous air pollutions in Beijing are directly related to the airflow transport from outside of Beijing (An et al., 2007; Yang et al., 2010). In the present study, the horizontal transport flux intensity is defined as the horizontal wind speed on the grid border multiplied by the pollutants concentration of the corresponding grid from which the airflows comes (Jiang et al., 2008). Considering that trans-boundary transport mainly occurs within the PBL, the study also focuses on the contribution of trans-boundary transport of pollutants within PBL over Beijing and its surrounding areas. Previous studies have shown that the average mixing layer height is approximately between 600 - 800 m during summertime, with the maximum during noontime higher than 1000 m (Wang et al., 2015; Tang et al., 2016). Figure 9 shows the temporal variations of net horizontal transport flux of $PM_{2.5}$, $O_3$ and $NO_2$ through Beijing boundary and the pollutants contributions from non-Beijing emissions to the air quality in Beijing city. The hourly $PM_{2.5}$, $O_3$ and $NO_2$ contributions of non-Beijing*

*emissions generally have the same variation trend as the horizontal transport flux, indicating that the contribution of surrounding sources plays an important role in high pollutants concentrations in Beijing during the study episode. For example, the $O_3$ net flux also has the similar peak in the afternoon as the $O_3$ contribution from the non-Beijing emissions. As discussed in Section 3.1.3, the prevailing south wind dominates BTH, so the largest flux intensity are from the south, with the average of 103.3 g $s^{-1}$ and 244.5 g $s^{-1}$ for $PM_{2.5}$ and $O_3$, respectively (SI-Table 1), indicating that the pollutants are mainly from south. The average horizontal transport fluxes for $PM_{2.5}$ and $O_3$ during the episode are 68.2 g $s^{-1}$ and 68.5 g $s^{-1}$, respectively, showing important contributions of non-Beijing emissions to the air quality in Beijing.*"

**3 Comment:** For the purpose of this paper, the accuracy of the emission inventory is very important. However, the description of the emission inventory in the manuscript is too brief. What year does the emission inventory based upon? Is it for the year 2006 as in Zhang et al. (2009) or is it updated? What are emissions from Beijing compared with those from the surrounding regions? How uncertain is the emission inventory? This information is essential for a reader to assess the significance of the paper's conclusion.

**Response:** We have added a Figure (Figure 2) to show the spatial distribution of anthropogenic $NO_x$, $VOC_s$, OC, and $SO_2$ emission rates, and a table (Table 1) to compare the emissions of Beijing with its surrounding areas. We have clarified in section 2.2: *"The anthropogenic emissions are developed by Zhang et al. (2009), which is based on the 2013 emission inventory, including contributions from agriculture, industry, power generation, residential, and transportation sources. The $SO_2$, $NO_x$, and CO emissions have been adjusted according to their observed trends from 2013 to 2015 in the present study, but the VOCs emissions are not changed considering that the VOCs emissions are still not fully considered in the current air pollutant control strategy. The major pollutants emissions used in the model simulation for Beijing, Tianjin, and the neighboring provinces (Hebei, Shanxi, and Shandong) are summarized in Table 1. Obviously, high anthropogenic emissions are distributed outside of Beijing, especially in Hebei and Shandong provinces. Figure 2 presents distributions of the emission rates of VOCs, $NO_x$, OC, and $SO_2$ in the simulation domain,*

*showing that the anthropogenic emissions are generally concentrated in urban areas. It is worth noting that uncertainties of the emission inventory used in the study are still large taking into consideration the rapid changes in anthropogenic emissions that are not fully reflected in the current emission inventories, particularly since implementation of the APPCAP, and the complexity of pollutants precursors.”*

**4 Comment:** In addition, as the authors pointed out, the emissions have been greatly reduced in Beijing in recent years. Therefore, if emission inventory for multiple years were available, it would be very interesting to conduct additional simulations and calculations and see how the implementation of APPCAP affects the contributions of trans-boundary transport.

**Response:** We have conducted sensitivity simulations during the period from 5 to 14 July 2015 using the 2013 emission inventory. The detailed discussion has been included in the Supplementary Information Section SI-2:

*“**Section SI-2: Contributions of Trans-boundary Transport to $PM_{2.5}$ and $O_3$ Concentrations in Beijing from 5 to 14 July 2015 Based on 2013 EI***

*In order to investigate the effect of the APPCAP on the contribution of trans-boundary transport, sensitivity simulations for the air pollution event from 5 to 14 July 2015 have been performed using the 2013 emission inventory. SI-Table 3 shows the average $PM_{2.5}$ contribution in Beijing from only-Beijing emissions, non-Beijing emissions, emission interactions, and background. During the study episodes, the average $PM_{2.5}$ contribution in Beijing from only-Beijing emissions from 5 to 14 July 2015 is 13.2%, which is lower than that using 2015 emission inventory. The non-Beijing emissions and the emission interactions contribute 65.2% and 6.5% to the $PM_{2.5}$ level in Beijing, respectively. Therefore, the $PM_{2.5}$ contribution caused by the trans-boundary transport is about 71.7% of $PM_{2.5}$ concentration in Beijing, which is higher than that with 2015 emission inventory. The background $PM_{2.5}$ contribution to Beijing is 15.0%. SI-Table 4 gives the average $O_3$ contributions from 12:00 to 18:00 BJT in Beijing from only-Beijing emissions, non-Beijing emissions, emission interactions, and background. The only-Beijing emissions contribute about 23.6% on average to the afternoon $O_3$ level in Beijing, varying from 16.4% to 29.7%, slightly higher than that with 2015 emission inventory. The outside emissions contribute more than local sources, with*

*an average of 38.0%, and the emissions interactions also decrease the $O_3$ level by 5.5% on average. The background also plays an important role in the $O_3$ level in the afternoon when using the 2013 emission inventory, with an average contribution of 44.0%. The $O_3$ contribution caused by the trans-boundary transport of outside emissions is approximately 32.5% of the $O_3$ concentration, higher than that with 2015 emission inventory. Hence, in general, the $O_3$ and $PM_{2.5}$ contribution of outside emissions has gradually decreased since implementation of the APPCAP."*

**5 Comment:** The authors used the FSA method to separate the impact of local and non-local emissions. I have several questions about the results that the authors present.

**(a)** In Table 2, the background contribution $f_0$ varies from 32.6 ppbv to 62.9 ppbv. Why does background vary so much? In addition, $f_0$ and $f'_S$ (surrounding) anti-correlates very well ($R^2=0.89$ based on my calculation). In Table 3, $f_0$ and $f'_S$ also anti-correlates really well ($R^2=0.92$). Why is that? The author should give more discussion to provide insight into these interesting results.

**Response:** We have clarified in Section 3.2.2: *"The background $O_3$ contribution varies from 32.6% to 62.9% during the episode, which is primarily determined by the prevailing wind direction. When the northerly wind is prevalent, the clean airflow from the north affects Beijing, enhancing the background $O_3$ contribution, such as on 5, 13, and 14 July 2015. However, when the polluted airflow from the south impacts Beijing, the background $O_3$ contribution is decreased."*

In general, the airflow influencing Beijing includes the $O_3$ from outside emissions and background, so when the background $O_3$ or $PM_{2.5}$ contribution is low, the $O_3$ or $PM_{2.5}$ contribution of outside emissions is high, Therefore, $f_0$ and $f'_S$ (surrounding) anti-correlates very well.

**(b)** In Table 2 and Figure 10(a). The authors show that local contribution to $O_3$ is much less than non-local contributions. Beijing, with so much traffic, should have large amount of $NO_x$ emissions. Given $NO_x$ lifetime in the summer should be on the order of several hours,

regional transport of $NO_x$ should not be very significant. So the question follows, why is regional contribution so larger? Is the input of regional $O_3$ or input of precursors? If it's the input of $O_3$, why do regions surrounding Beijing have such high $O_3$ production? Do they emit a lot of $NO_x$ and $VOC_s$. The authors should discuss the matter by referring to emission inventory (See major comment 2) and flux diagnosis (See major comment 1).

**Response:** We have added a Table in the supplement (SI-Table 2), and also explained the role of the $O_3$ precursors by referring to the emission inventory and flux diagnosis according to the referee's suggestion. We have added a paragraph in section 3.2.2 to provide a detailed description of the role of $O_3$ precursors, and the reason why the regional transport is dominant: *"Previous studies have proposed that the regional transport of $O_3$ precursors can play an important role in inducing the high $O_3$ level in Beijing (Wang et al., 2009; Zhang et al., 2014). SI-Table 2 provides the average $NO_2$ contributions in Beijing from only-Beijing emissions, non-Beijing emissions, emission interactions, and background. Different from $O_3$, the local emissions dominate the level of $NO_2$ in Beijing area, with an average contribution of 70.3% during the study episode. The average contribution of non-Beijing emissions, emission interactions and background are 24.8%, 0.9% and 4.0%, respectively. The contribution of background to $O_3$ concentrations is obvious, which is much more different from that for $NO_2$. In addition, the trans-boundary transport flux of $NO_2$ is much lower than that of $O_3$ (Figure 9). Given $NO_x$ lifetime in the summer is short, regional transport of $NO_x$ is not important. Furthermore, the emissions of $NO_x$ and $VOC_s$ around Beijing are much more than those in Beijing, especially in Hebei and Shandong provinces, which is subject to contribute more $O_3$ production (Table 1). Compared to the direct input of regional $O_3$, the transport of $O_3$ precursors probably does not play an important role in the high $O_3$ level in Beijing."*

**(c)** In Table 3 and Figure 10(b), the authors show somewhat counter-intuitive results that non-local emissions almost always contribute more than 50% of Beijing and $f'_s$ follows the $PM_{2.5}$ concentration perfectly. In my opinion, the authors should show additional results using emission inventory, flux diagnosis, etc. to convince a reader that their calculation is right and consistent. Or, it may be a good idea for the authors to show simulation and FSA

results outside an episode in the same summer to give readers a sense of how the "control" case looks like.

**Response:** We have conducted another simulation from 22 to 28 May 2015 to clarify the significant contribution of trans-boundary transport to the air quality in Beijing. The detailed discussion can be seen in the Supplementary Information Section SI-3:

*"**Section SI-3: Contribution of Trans-boundary Transport to PM$_{2.5}$ and O$_3$ Concentrations in Beijing from 22 to 28 July 2015 based on 2015 EI***

*The model simulations and FSA results from 5 to 14 July 2015 show that the non-Beijing emissions play an important role in the air quality in Beijing. In order to further confirm the important role of trans-boundary transport, a severe air pollution episode from 22 to 28 May 2015 in NCP is simulated using the WRF-CHEM model. SI-Table 5 shows the average PM$_{2.5}$ contribution in Beijing from only-Beijing emissions, non-Beijing emissions, emission interactions, and background. The average contribution from only-Beijing emissions is 11.5%, while the average contribution from non-Beijing emissions is 62.4%, varying from 35.4% to 73.4%, which is much higher than the contribution from local source. The emission interactions also increase the PM$_{2.5}$ level in Beijing, with the average contribution of 6.3%, varying from 2.4% to 9.4%. The background contributes 17.6% to the PM$_{2.5}$ concentration in Beijing on average. Therefore, the PM$_{2.5}$ contribution induced by trans-boundary transport is about 68.7% of PM$_{2.5}$ level in Beijing during the episode from 22 to 28 May 2015, indicating the substantial impact of trans-boundary transport on the PM$_{2.5}$ concentration in Beijing. SI-Table 6 presents the average O$_3$ contributions from 12:00 to 18:00 BJT in Beijing from only-Beijing emissions, non-Beijing emissions, emission interactions, and background. The local sources contribute about 17.6% on average in the afternoon to the O$_3$ level in Beijing. The non-Beijing emissions contribute more than local emissions, with the average contribution of 42.4%, ranging from 16.9% to 51.0%. The emission interaction between only-Beijing emissions and non-Beijing emissions increase or decrease the O$_3$ level in Beijing in different time due to the nonlinear process of O$_3$ production, with contributions ranging from -2.5% to 2.1%, but the emission interactions in Beijing decrease the O$_3$ concentration by 0.4% on average. The background plays also an*

*important role in the afternoon O₃ concentration, with average contribution of 40.4%. The contribution from trans-boundary transport of the outside emissions is about 42.0% during the period from 22 to 28 May 2015, further indicating that the important role of trans-boundary transport in the O₃ level in Beijing."*

**(d)** More fundamentally, I am concerned that the FSA results in the paper are somewhat misleading to readers because in the four simulations to derive FSA, the author turn on/off the local or non-local emissions completely. As a result of the nonlinear chemistry, these simulations cannot give accurate information about the local sensitivity of air quality to emission reduction. Theoretically, it is possible that reducing local emissions may still be more effective than reducing non-local emissions. I suggest the authors to make clear in the text about the limitation of their method.

**Response:** We have clarified in Conclusion: *"The FSA method can calculate the individual and synergistic contribution of only-Beijing and non-Beijing emissions by including or excluding the local or non-local emissions. However, considering the nonlinear chemistry of PM₂.₅ and O₃, especially regarding O₃ formation, the method might not well provide how the air quality is accurately when taking different emission reduction measures. Therefore, in the future study, sensitivity simulations of different emission reduction measures are needed to design reasonable emission control strategies."*

**(e)** The paper also lacks sufficient discussion of the results in the context of previous studies. The authors mentioned several previous studies on local vs. non-local emissions. The result of this paper stands out as reporting most significant non-local contributions. The paper will be much better if the authors can discuss their paper in context of these studies (in terms of method, results, discrepancies, or agreement, etc.).

**Response:** We have included discussions of the results in context of previous studies in Section 3.2.2 as follows:

*"Using the ozone source apportionment technique, Wang et al., (2009) have emphasized that local emissions are the most important contributor to high O₃ levels from June to July in*

*2000 in Beijing urban area because the emissions rates there are significantly higher than the average level in the surrounding areas. Based on CMAQ simulations of Beijing, Streets et al., (2007) have estimated that 35%—60% of the high $O_3$ concentration at the Olympic Stadium site could be attributed to non-Beijing emissions, with Hebei Province contributing 20%—30% of Beijing's ozone concentrations during the prevailing south wind. Wang et al., (2008) have found that the average contribution from non-Beijing emissions to the $O_3$ levels in Beijing is 30.0% and the maximum of daily contributions is as high as 56.5% in August 2006."*

*"Previous studies have also demonstrated the dominant role of non-Beijing emission in the $PM_{2.5}$ level in Beijing. Based on CMAQ model, Streets et al., (2007) have reported that average contribution of regional transport to $PM_{2.5}$ at the Olympic Stadium can be 34%, up to 50%—70% under prevailing south winds. Guo et al. (2010) have provided a rough estimation that the regional transport can contribute 69% of the $PM_{10}$ and 87% of the $PM_{1.8}$ in Beijing local area using the short and low time resolution data in the summer. Combining the $PM_{2.5}$ observations and MM5-CMAQ model results, regional transport is estimated to contribute 54.6% of the $PM_{2.5}$ concentration during the polluted period, with an annual average $PM_{2.5}$ contribution of 42.4% (Lang et al., 2013). Using the long-term measurements of $PM_{2.5}$ mass concentrations from 2005 to 2010 at urban Beijing, and trajectory cluster and receptor models, the average contribution of long-distance transport to Beijing's $PM_{2.5}$ level can be approximately 75.2% in the summer (Wang et al., 2015)."*

**Minor comments**

**(1)** Abstract. The words "pure local emissions" and "pure emissions outside Beijing" are confusing. Are there impure emissions? I suggest using "local emissions", "non-local (or non-Beijing) emissions", and "interactions between local and non-local emissions".

**Response:** We have changed "pure local emissions" and "pure emissions outside of Beijing" to "only-Beijing emissions" and "non-Beijing emissions", as the referee suggested.

**(2)** Line 121. Is NCEP reanalysis only used for boundary and initial conditions? Is WRF configured to nudge the meteorology fields towards reanalysis?

**Response:** We have clarified in Section 2.2: *"The NCEP 1° × 1° reanalysis data are used to obtain the meteorological initial and boundary conditions, and the meteorological simulations are not nudged in the study."*

**(3)** Section 3.1. This section is only a summary of changes in air quality of Beijing during recent years, which are background information rather than research results. Therefore this section should not be within "results and discussion".

**Response:** The Section 3.1 has been moved into the Section 2.5 as the referee suggested.

**(4)** Line 217-219, 243-238, 290. The author attributes the model errors to inability of WRF-CHEM to resolve convection in several occasions. However, these statements are not backed by any data (e.g., observations of convective clouds etc.). Given there are so many plausible error sources in a 3-D chemical transport model, I would suggest not to make guesses on what leads to the model errors (emission errors, meteorology errors, etc.), unless supported by observation data or model sensitivity test.

**Response:** We have removed the sentences as the referee suggested. *"The underestimation of PM$_{2.5}$ concentrations on July 8 and overestimation on July 11 and 14 are still rather large, perhaps caused by the simulated wind field uncertainties that influence the pollutants transports from outside of Beijing or lack of resolving convective clouds due to the 6 km horizontal resolution. "* in the line 217-219, *"There are two possible reasons for the biases in sulfate simulations. Firstly, the model is not able to resolve well convective clouds due to the 6 km horizontal resolution used in simulations, reducing the sulfate production from cloud processes. Secondly, a large amount of SO$_2$ is released from point sources, such as power plants or agglomerated industrial zones, and the transport of SO$_2$ from point sources is much sensitive to wind field simulations. "* in the line 243-248, and the sentence *"possibly caused by the active convections in the afternoon, which cannot be well resolved in WRF-CHEM model. "* in the line 290.

**(5)** Line 250. The title number should be 3.2.3.

**Response:** We have changed the title number.

---

## Author Comment (AC3) · 17 Nov 2016

The comment was uploaded in the form of a supplement: http://www.atmos-chem-phys-discuss.net/acp-2016-705/acp-2016-705-AC3-supplement.pdf

---

## Author Comment (AC4) · 17 Nov 2016

**Reply to Anonymous Referee #3**

We thank the reviewer for the careful reading of the manuscript and helpful comments. We have revised the manuscript following the suggestion, as described below.

**General comments**

This manuscript presents a WRF-CHEM modeling study using to evaluate the contribution of regional emissions to the air quality in Beijing during a summertime pollution episode using the FSA analysis. This study comes timely as there have been debates over whether local emissions play the major contribution to the air pollution in Beijing. The methodology is sound, and the results are well presented and organized. I would recommend it for publishing with a few minor revisions.

**Specific comments**

**1 Comment:** P1 line 27-28 and p21 lines 462-463, it is a big jump to extrapolate the results from an episode to the whole summer season. You need to prove that the episode studied is representative of the summertime situation in Beijing. In addition, air quality"primarily determined by the trans-boundary transport" may only be applied to $PM_{2.5}$, not $O_3$, since background $O_3$ accounts for 46% of the afternoon $O_3$ (line 338). Maybe I have a misunderstanding in here, which is related to the comment below.

**Response:** The description of the study episode has been added in Section 2.2 as follows: *"The maximum of $O_3$ concentration is higher than 350 μg $m^{-3}$, and the maximum of $PM_{2.5}$ concentration can reach a high level exceeding 150 μg $m^{-3}$. SI-Figures 1a-c show the daily averages of the temperature, relative humidity, and wind speed in Beijing during the summer of 2015. The minimum air temperature is 18.7 $^oC$, and the maximum air temperature is 40 $^oC$ during the summer, with average of 25.7 $^oC$. The average relative humidity is 63.8%. The southeast or southwest wind is prevailing over NCP due to the influence of East Asian summer monsoon (Zhang et al., 2010), with the average wind speed of 5.6 m $s^{-1}$ in the summer of 2015. During the study period, the average temperature, relative humidity, and*

*wind speed are 28.4 $^{o}$C, 51.7% and 6.3 m s$^{-1}$, respectively, indicating typical summertime meteorological conditions. During the summer of 2015, the average PM$_{2.5}$ concentration is 56.1 μg m$^{-3}$ and the average O$_3$ concentration in the afternoon is 216.4 μg m$^{-3}$ (SI-Figures 1d-e). The high O$_3$ and PM$_{2.5}$ event occurs frequently during the summertime of 2015, so the study period can well represent the summertime O$_3$ and PM$_{2.5}$ pollution in Beijing, and provides a suitable case for observation analyses and model simulations to investigate the effect of trans-boundary transport on the summertime air quality of Beijing. "*

We have changed "primarily determined by the trans-boundary transport" to "*generally determined by the trans-boundary transport*", considering the important O$_3$ contribution of background in the abstract.

**2 Comment:** Confusions on some concepts. To my understanding, the "trans-boundary transport" term refers to the transport of regional anthropogenic emissions (i.e., $f_S$), and it does not include "background" ($f_0$). Am I right? In addition, in the FSA analysis, which term include the biogenic emissions? Are the interactions between anthropogenic and biogenic emissions accounted? Does the trans-boundary transport include the impacts of biogenic emissions?

**Response:** The "trans-boundary transport" defined in this study does not include the contribution of background ($f_0$). In the study, we have not differentiated the individual effect of anthropogenic and biogenic emissions, and the biogenic emissions have been regarded as background considering that the biogenic emissions provide natural O$_3$ precursors and cannot be anthropogenically controlled.

**3 Comment:** In both the abstract and summary sections, besides considering the uncertainties in emissions and meteorology, simulations for more pollution episodes should be addressed to "evaluate trans-boundary transport contributions to the air quality in Beijing for supporting the design and implementation of emission control strategies".

**Response:** We have added the sentence in the Conclusion: "*In addition, simulations for more pollution episodes should be investigated to evaluate the contribution of trans-boundary*

*contributions to the air quality in Beijing for supporting the design and implementation of emission control strategies.*".

**4 Comment:** P2-3, lines 55-56, "daily average of up to 110...", daily average or daytime average?

**Response:** According to Wang et al. (2016), the maximum $O_3$ concentration during the polluted episode in summer of 2014 can exceed 300 µg m$^{-3}$, so it should be "daily average" in lines 55-56.

**5 Comment:** P5 lines 111-113, more descriptions of the episode are needed, including meteorological conditions, which may help to add information whether or not the episode is representative of the summertime air pollution in Beijing. Also, what does the "mean daily" mean here? episode average?

**Response:** Please refer to the response of Comment 1. The "mean daily" can be interpreted as episode average.

**6 Comment:** P9 line 191-193, the differences in CO, $SO_2$, $NO_x$ and $PM_{2.5}$ between 2013 and 2015 are attributed solely to the emission change. Are the meteorological conditions similar between these two periods?

**Response:** We have clarified in Section 2.5: " *The rainy days during summertime in Beijing are 43 and 46 days in 2013 and 2015, respectively, showing the similar meteorological conditions between the two years. Therefore, in general, the air pollutants variations between 2013 and 2015 can be mainly attributed to implementation of the APPCAP.*"

**7 Comment:** P10-11 Section 3.2, $NO_3^-$ and $NH_4^+$ are shown in Figure 3, but there are no discussions or descriptions of these two components. Point comparisons may also contribute to the biases.

**Response:** We have added description of $NO_3^-$ and $NH_4^+$ as the referee suggested in Section 3.1.2 as follows: *"The model reasonably well reproduces the observed temporal*

*variations of SOA, nitrate, and ammonium, with IOAs exceeding 0.75. The model also replicates well the peak concentrations of SOA, nitrate and ammonium at the rush hour, but generally underestimates the measured SOA, nitrate, and ammonium concentrations, with MBs of -1.1 μg m$^{-3}$, -0.7 μg m$^{-3}$, and -0.5 μg m$^{-3}$, respectively. For nitrate and ammonium aerosols, the underestimation occurs mainly on 8 July 2015."*

**8 Comment:** Figures 10 and 11, regarding the contributions from total emissions, emissions from Beijing, and emissions outside Beijing. As I understand, the last two terms are calculated as $f'_B + f'_{BS}$ (or $f_{BS} - f_S$) and $f'_S + f'_{BS}$ (or $f_{BS} - f_B$), but how is the first term (contribution from total emissions) calculated? $f_{BS} - f_0$? or sum of the last two terms? Captions in Fig. 10 and 11 are a bit of confusing: $f_B$, $f_S$, and $f_{BS}$ represent simulation results, not contributions.

**Response:** We performed four simulations in this study: (1) $f_{BS}$ with all the emissions; (2) $f_B$ with Beijing emissions alone; (3) $f_S$ with non-Beijing emissions alone; (4) $f_0$ without all the anthropogenic emissions. Therefore, the contribution of only-Beijing emissions is represented as $f_B - f_0$ ($f'_B$), and the contribution of the non-Beijing emissions is represented as $f_S - f_0$ ($f'_S$). The pollutant level in Beijing is determined by the contribution of only-Beijing emissions ($f'_B$), the trans-boundary transport of non-Beijing emissions ($f'_S$), emission interactions ($f'_{BS}, f_{BS} - f_B - f_S + f_0$), and background ($f_0$).

We have changed the captions of Figures 10: *"Temporal variations of the average near-surface (a) O$_3$ and (b) PM$_{2.5}$ concentrations from $f_{BS}$ with all the emissions (black line), $f_B$ with Beijing emissions alone (blue line), and $f_S$ with non-Beijing emissions alone (red line) in Beijing from 5 to 14 July 2015."*

**9 Comment:** P15 line 337, 45.6%, but in table 2, the number is 46.1%.

**Response:** We have changed "45.6%" to "46.1%".

**10 Comment:** Table 3 is shown and is not discussed.

**Response:** We have changed the original Table 3 to Table 4 in Section 3.2.2 in the present

study. We have discussed as follows: "*Table 4 shows the average $PM_{2.5}$ contribution in Beijing from only-Beijing emissions, non-Beijing emissions, emission interactions, and background. During the study episode, the average $PM_{2.5}$ contribution from local emissions is 13.7%, which is much lower than the contribution of 61.5% from the emissions outside of Beijing, further showing the dominant role of the trans-boundary transport in the Beijing $PM_{2.5}$ pollution. The emission interactions enhance the $PM_{2.5}$ level in Beijing on average, with a contribution of 5.9%. The background $PM_{2.5}$ contribution to Beijing is 18.9% on average, lower than those for $O_3$. The $PM_{2.5}$ contribution caused by the trans-boundary transport is about 67.4% of $PM_{2.5}$ concentrations in Beijing, indicating that the cooperation with neighboring provinces to control the $PM_{2.5}$ level is a key for Beijing to improve air quality. Previous studies have also demonstrated the dominant role of non-Beijing emission in the $PM_{2.5}$ level in Beijing. Based on CMAQ model, Streets et al., (2007) have reported that average contribution of regional transport to $PM_{2.5}$ at the Olympic Stadium can be 34%, up to 50%—70% under prevailing south winds. Guo et al. (2010) have provided a rough estimation that the regional transport can contribute 69% of the $PM_{10}$ and 87% of the $PM_{1.8}$ in Beijing local area using the short and low time resolution data in the summer. Combining the $PM_{2.5}$ observations and MM5-CMAQ model results, regional transport is estimated to contribute 54.6% of the $PM_{2.5}$ concentration during the polluted period, with an annual average $PM_{2.5}$ contribution of 42.4% (Lang et al., 2013). Using the long-term measurements of $PM_{2.5}$ mass concentrations from 2005 to 2010 at urban Beijing, and trajectory cluster and receptor models, the average contribution of long-distance transport to Beijing's $PM_{2.5}$ level can be approximately 75.2% in the summer (Wang et al., 2015)*".

**Technical corrections**

1 Line 25, the word "more" has been replaced by "higher".

2 Line 32, "reasonably" has been changed to "better".

3 Line 134, Stein et al. (1993) has been revised as Stein and Alpert (1993).

4 Line 189, the word "hourly" has been deleted.

5 Line 227, "the failure of " has changed to "biased".

6 Lines 533-538, the two references have been added in the manuscript.

---

## Editor Decision (ED1)

In my judgment the paper is suitable for publication when some further minor revisions are made.

**I.** In accord with Review #1, I think that some of the discussion is overly speculative, and thus does not add to the paper. Generally, removal of some material, rather than additional discussion is required. Specific suggestions follow.

Lines 422-424: I suggest eliminating the last sentence. It may well be correct, but that correctness is not demonstrated in this paper, so it can simply be removed.

Lines 477-479: I suggest eliminating the phrase in yellow, i.e., ", which is due to the increased atmospheric oxidation capability caused by elevated $O_3$ concentrations during summertime." I do not believe that it is well established that elevated $O_3$ concentrations cause high SOA. Rather I think that both high $O_3$ and high SOA are products of the oxidation processes.

Line 505: Should be: "... condensable gases do not change, more organic condensable gases partition into the ..."

Line 514: eliminate " deliberate"

Lines 515-528: These sentences give plausible explanations for the interactions determined from the analysis, but the explanations are speculative. These sentences should be greatly shortened, limited to reporting the magnitude of the emission interactions.

Lines 562-575: This discussion is also largely speculative, and in some instances, may not be correct. For example, Mexico City does have substantial industrial emissions. It may be useful to point out that a comprehensive model comparison of summertime pollution in Mexico City and Beijing could be illuminating, but the present discussion is too speculative; please shorten.

Lines 576-589: This material has been added in response to a Reviewer's comment, but I do not think that it is well done. It would be improved if shortened to something like:
"This study mainly aims at providing a quantification of the effect of trans-boundary transport on the air quality in Beijing. It demonstrates that the effective approach to improve air quality in Beijing is to reduce both local and non-Beijing emissions in BTH. Further sensitivity simulations of different emission reduction measures are needed to design the most efficient emission control strategies."
Further, even this paragraph is somewhat duplicative of the final paragraph, so the best approach is to combine the final two paragraphs of the revised manuscript into a single paragraph that concisely and clearly combines the short paragraph suggested above with the paragraph on lines 590-600.

**II.** In general I think that the English usage in this paper is quite good. However, in many places there is confusion of present and past tenses and singular and plural. Some examples from the Introduction and the Summary and Conclusions Sections, with suggested changes indicated in yellow are given below. I suggest that the paper be edited thoroughout for consistent use of these issues.

Line 53: "summertime O3 mass concentrations reached high levels in 2014 in Beijing,"

Line 55: " maximum daily O3 concentrations were higher than 150 μg m-3 during the summer in 2015"

Line 60: " daytime average O3 concentration still increased rapidly (Tang et al., 2009;"

Line 74: " the transport from the environs of Beijing contributed about 55%"

Line 77: " (hereafter referred to as APPCAP) has been implemented, which was released"

Line 532: " and PM2.5 is simulated using ..."

Line 535: " concentration in the afternoon has increased by ..."  (been eliminated)

Line 536: " and Beijing still has experienced high O3 and/or PM2.5 concentrations frequently during"

Line 547: " included in model simulations, the O3 and PM2.5 concentrations in Beijing still remain high"

Line 548: Eliminate "levels"

---

## Author Response (AR2)

December 27, 2016

Dear Editor,

We have received the comments from the reviewer #1 of the manuscript. Below are our responses and the revisions that we have made in the manuscript.

Thank you for your efforts on this manuscript. We look forward to hearing from you.

Best Regards,

Guohui Li

**Reply to Anonymous Referee #1**

We thank the reviewer for the careful reading of the manuscript and helpful comments. We have revised the manuscript following the suggestion, as described below.

The manuscript has been improved from the last version. However, I still think it has not yet met the standard of an ACP paper.

**Major comments**

**1. Comment:** My major concern is the writing style of the paper. The authors tend to provide too much information and do not organize it well, which makes a reader difficult to follow.
An example is Line 506-534, where the authors try to explain a lot of things in a very long paragraph without a clear reasoning flow. In addition, Section 3.2.2 is 7 pages long and the topic of the section changes from $O_3$ to $PM_{2.5}$ and then to aerosol composition without indications of topic changes.

**Response:** We have divided the paragraph from Line 506-534 into two parts, including the explanations for the contributions of emission interactions to organic and inorganic aerosols, respectively. We have added Section 3.2.3 *"Trans-boundary transport contributions to PM$_{2.5}$ in Beijing"* and Section 3.2.4 *"Trans-boundary transport contributions to aerosol species in Beijing"* in the manuscript, and revised Section 3.2.2 as *"Trans-boundary transport contributions to O$_3$ in Beijing"*.

**Comment:** Line 408-417 is yet another example. The authors described several previous studies one by one with some unnecessary details. In my opinion, this information should be presented in a more concise way and should be put in the context of the current study.

**Response**: We have added the sentence in Section 3.2.2 as follows: *"The O$_3$ contributions in Beijing induced by the trans-boundary transport of non-Beijing emissions is about 31.5% of the O$_3$ concentration during the study episodes, which is in agreement with previous studies (Streets et al., 2007; Wang et al., 2008), indicating that the trans-boundary transport constitutes the*

*main reason for the elevated O₃ level in Beijing after implementation of the APPCAP.”*

**Comment**: In addition, some wording may not be right, for example, "pure contribution" (L207), "pure impact" (L211), "outside emissions" (L24, 26, 375, 390, 509, 560), and "only-Beijing emissions" (throughout the text).

**Response:** We have revised the wording in the manuscript as suggested.

**2. Comment:** I am also concerned about how a reader would interpret the policy implication of the study because the authors did not thoroughly discuss the matter in the text. I think the authors need some extra work on the discussion, especially on how their work adds to the debate on whether local or non-local emissions play the major contribution to the air pollution in Beijing (Guo et al., 2014; Li et al., 2015; Zhang et al., 2015).

**Response:** We have added a further discussion in Conclusion to explain the role of non-Beijing emissions as follows: *"However, it is still controversial on whether local or non-local emissions play a dominant role in the air quality in Beijing (Guo et al., 2010, 2014; Li et al., 2015; Zhang et al., 2015). When only considering the local emissions, Beijing only experiences O₃ pollution, and the PM$_{2.5}$ level is low during summertime, which is comparable to the air quality in Mexico City. Mexico City has once been one of the most polluted cities in the world, but the air quality has been greatly improved in recent years after taking emission control strategies (Molina et al., 2002, 2007, 2010). Beijing and Mexico City now have similar emission sources, including transportation and residential living, but Beijing is surrounded by the highly industrialized areas in the south and east. When considering the trans-boundary transport of the pollutants from non-Beijing emissions, the O₃ and PM$_{2.5}$ levels in Beijing are remarkably increased, much higher than those in Mexico City, showing the important role of trans-boundary transport in the air quality in Beijing. Hence, the cooperation with neighboring provinces to decrease pollutant emissions is the optimum approach to mitigate the air pollution in Beijing.”*

The paper may lead a reader to conclude that the major culprit for air pollution in Beijing is neighboring regions. However, this interpretation may not be completely right.

**(1) Comment:** As shown in Figure 2, the emission rate from neighboring regions is not significantly higher than Beijing. Although the total emissions (Table 1) from the neighboring regions are large, the areas of those regions are also very large. From Figure 2, the area average emission rate (numbers in Table 1 divided by areas) of Beijing is probably still highest.

**Response:** We have clarified in Section 2.2: "*As shown in Figure 2, the total emissions from neighboring regions are much more than those in Beijing, and the emission rates in Tianjin, the south of Hebei and Shandong are also higher than those in Beijing, particularly with regard to $SO_2$ emissions. Therefore, when the south or east wind is prevailing in NCP, the severe air pollution can be formed in Beijing when precursor emissions in highly industrialized areas chemically react as they are carried toward Beijing, blocked by mountains and further accumulated and interacted with those in Beijing.*"

**(2) Comment:** Consistent with (1), Figure 9 and SI-Table 1 shows that the flux is almost symmetric around 0, suggesting Beijing is likely to contribute equal amount of pollution to the neighboring regions.

**Response:** We have clarified in the Section 3.2.1: "*As discussed in Section 3.1.3, the prevailing south wind dominates in BTH, so the largest flux intensity are from the south, with the average of 103.3 $g$ $s^{-1}$ and 244.5 $g$ $s^{-1}$ for $PM_{2.5}$ and $O_3$, respectively (SI-Table 1), indicating that the pollutants are mainly from the south. It should be noted that the flux of $O_3$ is mainly focused on the afternoon from 12:00 to 18:00 BJT. The average net horizontal transport fluxes for $PM_{2.5}$ and $O_3$ during the episode are 68.2 $g$ $s^{-1}$ and 68.5 $g$ $s^{-1}$, respectively, showing important contributions of non-Beijing emissions to the air quality in Beijing.*"

**(3) Comment:** BTH is a polluted air basin (Zhao et al., 2009; Parrish et al., 2015). Applying FSA to any city in BTH may generate similar results as in this paper.
The paper may also lead a reader to conclude that the most effective way to control air pollution in Beijing is to reduce non-Beijing emissions in BTH. The FSA method is based on simulations completely turn on/off emissions from a certain region. Because a) the method cannot provide information on the local sensitivities of air pollution to emission reduction and b) emission

reduction to zero in a vast region is apparently an infeasible scenario, inference of control strategies from the results is improper. In Line 568-575, the authors briefly mentioned this but this limitation is not explicitly stated.

**Response:** We have clarified the limitation of FSA method in Conclusion as follows: "*BTH has been considered as a polluted air basin (Zhao et al., 2009; Parrish et al., 2015). However, although Beijing has implemented aggressive emission control strategies, it still experiences $O_3$ and $PM_{2.5}$ pollutions during summertime, showing that the effective way to improve air quality in Beijing is to reduce non-Beijing emissions in BTH. The FSA method is based on simulations in which emissions from a certain region are completely turned on/off, which can calculate the individual and synergistic contribution of local Beijing and non-Beijing emissions by including or excluding the local or non-local emissions in this study. However, considering the nonlinear chemistry of $PM_{2.5}$ and $O_3$, especially regarding $O_3$ formation, the method might not well provide how the air quality is accurately when taking different emission reduction measures, and also emission reduction to zero in a vast region is apparently an infeasible scenario. This study mainly aims at providing a quantification of the effect of trans-boundary transport on the air quality in Beijing. Therefore, in the future study, sensitivity simulations of different emission reduction measures are needed to design reasonable emission control strategies.*"

**Minor comments**

**1 Comment:** Line 128 "2.2 Model Configuration"=> 2.2 Pollution episode simulation.

**Response:** We have revised the section title as "*2.2 Pollution Episode Simulation*".

**2 Comment:** Line 219: "2.4 Statistical Methods for Comparisons". => Statistical metrics for observation-model comparisons

**Response:** We have revised the section title as "*2.4 Statistical Metrics for Observation-Model Comparisons*".

**3 Comment**: Line 230: "2.5 Pollutants Measurements" =>2.5 Pollutant Measurements

**Response:** We have revised the section title as *"2.5 Pollutant Measurements"*.

**4 Comment:** Line 379: Apparently, …

**Response:** We have changed the " Therefore" to "Apparently" in Section 3.2.2.

**5 Comment:** Line 419: inducing the high $O_3$ concentrations level in Beijing

**Response:** We have revised the sentence as *"play an important role in inducing the high $O_3$ concentrations level in Beijing"* in Section 3.2.2.

**6 Comment:** Line 424-431: "The contribution of background to $O_3$ is obvious, which is much more different from that for $NO_2$. In addition, the trans-boundary transport flux of $NO_2$ is much lower than $O_3$ (Figure 9). Given NOx lifetime in the summer is short, regional transport of NOx is not important. Furthermore, the emissions of NOx and VOCs around Beijing are much more than those in Beijing, especially in Hebei and Shandong provinces, which is subject to contribute more $O_3$ production (Table 1). Compared to the direct input of regional $O_3$, the transport of $O_3$ precursors probably does not play an important role in the high $O_3$ level in Beijing."

Too much redundant or irrelevant sentences. I would change to "Compared to the direct input of regional O3, the regional transport of NOx is unlikely a significant contributor to high $O_3$ concentrations in Beijing, partly due to its short lifetime in the summer."

**Response:** We have changed the sentences as suggested in Section 3.2.2: *"Compared to the direct input of regional $O_3$, the regional transport of $NO_x$ is unlikely a significant contributor to high $O_3$ concentrations in Beijing, partly due to its short lifetime in the summer."*

**7 Comment:** Line 464: e.g., i.e.,

**Response:** We have revised it in the manuscript.

**8 Comment:** Line 483: "high atmospheric oxidation capability caused by elevated $O_3$ concentrations during summertime". High atmospheric oxidation capacity is not directly caused by high $O_3$ concentrations.

**Response:** We have revised the sentence as "*due to the increased atmospheric oxidation capability caused by elevated $O_3$ concentrations during summertime.*"

**9 Comment:** Line 489-491: Replicate of line 482-483.

**Response:** We have revised the sentence in Section 3.2.4 as follows: "*Secondary aerosol species dominate the $PM_{2.5}$ mass concentration in Beijing, with a contribution of 77.9%.*"

**10 Comment:** Line 500-501: "…which is caused by the aerosol radiative effect. It is clear that the PBL-pollution interaction plays an important role in the pollutant accumulation in Beijing…" This effect may not be called the aerosol radiative effect. I think PBL-pollution interaction is more proper.

**Response:** We have revised the "*the aerosol radiative effect*" as "*PBL-pollution interaction*" in Section 3.2.3 and Conclusion.

**11 Comment:** Section SI-3: My suggestion of a "control" case in last review was to present some results in the clean period, which the authors seemed to misunderstand. I was thinking a comparison between a clean and a polluted episode may bring some insight into the regional contribution. Section SI-3 presents a polluted episode similar to the one in the main text. The results are essentially the same. I would suggest removing this section and relevant sentences in the main text.

**Response:** We have removed this section in the main text. According to previous studies and the analysis of meteorological conditions, when the north wind is prevailing in BTH, the air quality

in Beijing is good due to the much more clean air transported from the north of China. In addition, we have performed analysis of the weather in Beijing during the summertime of 2015. There are 46 rainy days in Beijing, but the $O_3$ exceedance days with $O_3$ concentration more than 200 μg m$^{-3}$ are 53 days. So it is difficult to find a clean episode in Beijing without precipitation.

**References**

Parrish DD, Stockwell WR (2015) Urbanization and air pollution: Then and now. EOS 96(1):12–15.

Zhao, C., Y. Wang, and T. Zeng (2009) East China plains: A "basin" of ozone pollution, Environ. Sci. Technol., 43, 1911–1915.

Guo S, Hu M, Zamora ML, et al. Elucidating severe urban haze formation in China. Proceedings of the National Academy of Sciences of the United States of America. 2014;111(49):17373-17378. doi:10.1073/pnas.1419604111.

Li P, Yan R, Yu S, Wang S, Liu W, Bao H. Reinstate regional transport of $PM_{2.5}$ as a major cause of severe haze in Beijing. Proceedings of the National Academy of Sciences of the United States of America. 2015;112(21):E2739-E2740. doi:10.1073/pnas.1502596112.

Zhang R, Guo S, Levy Zamora M, Hu M. Reply to Li et al.: Insufficient evidence for the contribution of regional transport to severe haze formation in Beijing. Proceedings of the National Academy of Sciences of the United States of America. 2015;112(21):E2741. doi:10.1073/pnas.1503855112.

---

## Author Response (AR3)

January 14, 2017

Dear Dr. David Parrish,

We have received your comments of the manuscript. Below are our responses and the revisions that we have made in the manuscript.

Thank you for your efforts on this manuscript. We look forward to hearing from you.

Best Regards,

Guohui Li

**Reply to the Editor, Dr. David Parrish**

We thank the Editor, Dr. David Parrish for the careful reading of the manuscript and helpful comments. We have revised the manuscript following the suggestion, as described below.

In my judgment the paper is suitable for publication when some further minor revisions are made.

**I.** In accord with Review #1, I think that some of the discussion is overly speculative, and thus does not add to the paper. Generally, removal of some material, rather than additional discussion is required. Specific suggestions follow.

**Comment:** Lines 422-424: I suggest eliminating the last sentence. It may well be correct, but that correctness is not demonstrated in this paper, so it can simply be removed.

**Response:** We have eliminated the sentence in the manuscript in Section 3.2.2.

**Comment:** Lines 477-479: I suggest eliminating the phrase in yellow, i.e., ", which is due to the increased atmospheric oxidation capability caused by elevated $O_3$ concentrations during summertime." I do not believe that it is well established that elevated $O_3$ concentrations cause high SOA. Rather I think that both high $O_3$ and high SOA are products of the oxidation processes.

**Response:** We have deleted the sentence in the manuscript.

**Comment:** Line 505: Should be: "... condensable gases do not change, more organic condensable gases partition into the ..."

**Response:** We have changed the word "*participate*" in the manuscript as "*partition*".

**Comment:** Line 514: eliminate " deliberate"

**Response:** We have eliminated the word "*deliberate*".

**Comment:** Lines 515-528: These sentences give plausible explanations for the interactions determined from the analysis, but the explanations are speculative. These

sentences should be greatly shortened, limited to reporting the magnitude of the emission interactions.

**Response:** We have revised the sentence in Section 3.2.4 as follows: "*The nitrate contributions from emission interactions are 18.1%, much more than those for other aerosol constituent. The sulfate contribution from emission interactions is not significant, only 3.4%. The ammonium contributions from emissions interactions are 1.5%, similar to those of primary aerosol species.*"

**Comment:** Lines 562-575: This discussion is also largely speculative, and in some instances, may not be correct. For example, Mexico City does have substantial industrial emissions. It may be useful to point out that a comprehensive model comparison of summertime pollution in Mexico City and Beijing could be illuminating, but the present discussion is too speculative; please shorten.

**Response:** We have classified the discussion as follows: "*However, it is still controversial on whether local or non-local emissions play a dominant role in the air quality in Beijing (Guo et al., 2010, 2014; Li et al., 2015; Zhang et al., 2015). When only considering the local emissions, the summertime $PM_{2.5}$ level in Beijing is comparable to that in Mexico City. Mexico City has once been one of the most polluted cities in the world, but the air quality has been greatly improved in recent years after taking emission control strategies (Molina et al., 2002, 2007, 2010). Therefore, a comprehensive model comparison of summertime pollution in Mexico City and Beijing would be illuminating for elucidation of the contributions of trans-boundary transport to the air quality in Beijing.*"

**Comment:** Lines 576-589: This material has been added in response to a Reviewer's comment, but I do not think that it is well done. It would be improved if shortened to something like: "This study mainly aims at providing a quantification of the effect of trans-boundary transport on the air quality in Beijing. It demonstrates that the effective approach to improve air quality in Beijing is to reduce both local and non-Beijing emissions in BTH. Further sensitivity simulations of different emission reduction measures are needed to design the most efficient emission control strategies."

Further, even this paragraph is somewhat duplicative of the final paragraph, so the best approach is to combine the final two paragraphs of the revised manuscript into a single paragraph that concisely and clearly combines the short paragraph suggested above with the paragraph on lines 590-600.

**Response:** We have revised the paragraph in conclusion as follows: "*It is worth noting that, although the WRF-CHEM model well captures the spatial distributions and temporal variations of pollutants, the model biases still exist. The discrepancies between the predictions and observations are possibly caused by the uncertainties in the emission inventory and the meteorological fields simulations (Zhang et al., 2015). BTH has been considered as a polluted air basin (Zhao et al., 2009; Parrish et al., 2015), which frequently experience $O_3$ and $PM_{2.5}$ pollutions during summertime. Future studies need to be conducted to improve the WRF-CHEM model simulations, and further to assess the contributions of trans-boundary transport of emissions outside of Beijing to the air quality in Beijing, considering the rapid changes in anthropogenic emissions since implementation of the APPCAP. This study mainly aims at providing a quantification of the effect of trans-boundary transport on the air quality in Beijing. It demonstrates that the effective approach to improve air quality in Beijing is to reduce both local and non-Beijing emissions in BTH. Further sensitivity simulations of different emission reduction measures are needed to design the most efficient emission control strategies.*"

II. In general I think that the English usage in this paper is quite good. However, in many places there is confusion of present and past tenses and singular and plural. Some examples from the Introduction and the Summary and Conclusions Sections, with suggested changes indicated in yellow are given below. I suggest that the paper be edited throughout for consistent use of these issues.

**Comment:** Line 53: " summertime $O_3$ mass concentrations reached high levels in 2014 in Beijing,"

Response: We have revised it in the manuscript.

**Comment:** Line 55: " maximum daily $O_3$ concentrations were higher than 150 μg m$^{-3}$ during the summer in 2015"

**Response:** We have revised it in the manuscript.

**Comment:** Line 60: " daytime average $O_3$ concentration still increased rapidly (Tang et al., 2009;"

**Response:** We have revised it in the manuscript.

**Comment**: Line 74: " the transport from the environs of Beijing contributed about 55%"

**Response:** We have revised it in the manuscript.

**Comment:** Line 77: " (hereafter referred to as APPCAP) has been implemented, which was released"

**Response:** We have revised it in the manuscript.

**Comment:** Line 532: " and $PM_{2.5}$ is simulated using ..."

**Response:** We have revised it in the manuscript.

**Comment:** Line 535: " concentration in the afternoon has increased by ..." (been eliminated)

**Response:** We have eliminated "*been*" in the conclusion.

**Comment:** Line 536: " and Beijing still has experienced high $O_3$ and/or $PM_{2.5}$ concentrations frequently during"

**Response** : We have changed the word "*pollutions*" as "*concentrations*" in the conclusion.

**Comment:** Line 547: " included in model simulations, the $O_3$ and $PM_{2.5}$ concentrations in Beijing still remain high"

**Response:** We have changed the word "*considered*" as "*included*" in the manuscript.

**Comment:** Line 548: Eliminate " levels"

Response: We have eliminated "*levels*" in the conclusion.